

# A robust Upwind Mixed Hybrid Finite Element method for transport in variably saturated porous media

6
7

Anis Younes[1*], Hussein Hoteit[2], Rainer Helmig[3], Marwan Fahs[1]

[1] Institut Terre et Environnement de Strasbourg, Université de Strasbourg, CNRS, ENGEES, UMR 7063, 67084 Strasbourg, France

[2] Physical Science and Engineering Division, King Abdullah University of Science and Technology (KAUST), Thuwal, Saudi Arabia

[3] Institute for Modelling Hydraulic and Environmental Systems, University of Stuttgart, Pfaffenwaldring 61, 70569 Stuttgart, Germany


*Submitted to Hydrology and Earth System Sciences (HESS)*

Contact author: Anis Younes

E-mail: younes@unistra.fr



**Abstract**
The Mixed Finite Element (MFE) method is well adapted for the simulation of fluid flow in
heterogeneous porous media. However, when employed for the transport equation, it can
generate solutions with strong unphysical oscillations because of the hyperbolic nature of
advection. In this work, a robust upwind MFE scheme is proposed to avoid such unphysical
oscillations. The new scheme is a combination of the upwind edge/face centred finite volume
method with the hybrid formulation of the MFE method. The scheme ensures continuity of
both advective and dispersive fluxes between adjacent elements and allows to maintain the
time derivative continuous, which permits employment of high order time integration
methods via the Method of Lines (MOL).
Numerical simulations are performed in both saturated and unsaturated porous media to
investigate the robustness of the new upwind-MFE scheme. Results show that, contrarily to
the standard scheme, the upwind-MFE method generates stable solutions without under and
overshoots. The simulation of contaminant transport into a variably saturated porous medium
highlights the robustness of the proposed upwind scheme when combined with the MOL for
solving nonlinear problems.

*Keywords*:
Hybrid Mixed Finite Element, upwind scheme, advection-dispersion transport, numerical
oscillations, Method of Lines.


## 1. Introduction

The Mixed Finite Element (MFE) method (Raviart and Thomas, 1977; Brezzi *et al.*, 1985; Chavent and Jaffré, 1986; Brezzi and Fortin, 1991, Younes *et al.*, 2010) is known to be a robust numerical scheme for solving elliptic diffusion problems such as the fluid flow in heterogeneous porous media. The method combines advantages of the finite volumes, by ensuring local mass conservation and continuity of fluxes between adjacent cells, and advantages of finite elements by easily handling heterogeneous domains with discontinuous parameter distributions and unstructured meshes. As a consequence, the MFE method has been largely used for flow in porous media (see, for instance, the review of Younes *et al.* (2010) and references therein). The hybridization technique has been largely used with the MFE method to improve its efficiency (Chavent and Roberts, 1991; Traverso *et al.* 2013). This technique allows to reduce the total number of unknowns and produces a final system with a symmetric positive definite matrix. The unknowns with the hybrid-MFE method are the Lagrange multipliers which correspond to the traces of the scalar variable at edges/faces (Chavent and Jaffré, 1986).

When applied to transient diffusion equations with small time steps, the hybrid-MFE method can produce solutions with small unphysical over and undershoots (Hoteit *et al.*, 2002a, 2002b; Mazzia, 2008). A lumped formulation of the hybrid-MFE method was developed by Younes *et al.* (2006) to improve its monotonicity and reduce nonphysical oscillations. The lumped formulation ensures that the maximum principle is respected for parabolic diffusion equations on acute triangulations (Younes *et al.*, 2006). For more general 2D and 3D element shapes, the lumping procedure allows to significantly improve the monotonous character of the hybrid-MFE solution (Younes *et al.*, 2006; Koohbor *et al.*, 2020). As an illustration, the lumped formulation was shown to be more efficient and more robust than the standard hybrid formulation for the simulation of the challenging nonlinear problem of water infiltration into

an initially dry soil (Belfort *et al*., 2009). The lumped formulation has recently been used for
flow discretization in the case of density driven flow in saturated-unsaturated porous media
(Younes *et al*., 2022a).
However, the MFE method remains little used for the discretization of the full transport
equation. When employed to the advection-dispersion equation, the MFE method can
generate solutions with strong numerical instabilities in the case of advection-dominated
transport because of the hyperbolic nature of the advection operator. To avoid these
instabilities, one of the most popular and easiest ways is to use an upwind scheme. Indeed,
although upwind schemes introduce some numerical diffusion leading to an artificial
smearing of the numerical solution, they avoid unphysical oscillations and remain useful,
especially for large domains and regional field simulations. In the literature, some upwind
mixed finite element schemes have been employed to improve the robustness of the MFE
method for advection-dominated problems (Dawson, 1998; Dawson and Aizinger, 1999;
Radu *et al*., 2011; Vohralik, 2007; Brunner *et al*., 2014).
The main idea of an upwind scheme for an element $E$, is to calculate the mass flux exchanged
with its adjacent element $E$' using the concentration from $E$ in the case of an outflow and the
concentration from $E'$ in the case of an inflow. However, this idea cannot be applied as such
with the hybrid-MFE method since the hybridization procedure requires to express the flux at
the element interface as only a function of variables at the element $E$ (and not $E$'). To
overcome this difficulty, Radu *et al.* (2011), and Brunner *et al.* (2014) proposed an upwind
MFE method where, in the case of an inflow, the concentration at the adjacent element $E$' is
replaced by an approximation using the concentration at $E$ and the trace of concentration at
the interface $\partial_{EE'}$ by assuming that the edge concentration is the mean of the concentrations in
$E$ and $E$'. However, this assumption cannot be verified for a general configuration.
Furthermore, with such an assumption, each of the advective and dispersive fluxes is
discontinuous at the element interfaces, and continuity is only fulfilled for the total flux.
In this work, a new upwind-MFE method is proposed for solving the full transport equation
without requiring any approximation of the upwind concentration. The new scheme is a
combination of the upwind edge/face centered finite volume (FV) scheme with the lumped
formulation of the MFE method. It guarantees continuity of both advective and dispersive
fluxes at element interfaces. Further, the new upwind-MFE scheme maintains the time
derivative continuous and thus, allows to employ high order time integration methods via the
method of lines (MOL), which was shown to be very efficient for solving nonlinear problems
(see, for instance, Fahs *et al.* (2009) and Younes *et al.* (2009)).
This article is structured as follows. In section 2, we recall the hybrid-MFE method for the
discretization of the transport equation. In section 3, we introduce the new upwind-MFE
method based on the combination of the upwind edge/face FV scheme with the lumped
formulation of the MFE method. In section 4, numerical experiments are performed for
transport in saturated and unsaturated porous media to investigate the robustness of the new
developed upwind-MFE scheme. Some conclusions are given in the last section of the article.

## 126    2. The hybrid-MFE method for the advection-dispersion equation

The mass conservation of the contaminant in variably saturated porous media is:
$$\frac{\partial(\theta C)}{\partial t} + \nabla.\left(\tilde{\boldsymbol{q}}_a + \tilde{\boldsymbol{q}}_d\right) = 0 \qquad (1)$$
where $C$ is the normalized concentration [-], $\theta$ is water content [$L^3 L^{-3}$], ], $t$ is time [T],
$\tilde{\boldsymbol{q}}_a = \boldsymbol{q}C$ is the advective flux with $\boldsymbol{q}$ the Darcy velocity [$LT^{-1}$] and $\tilde{\boldsymbol{q}}_d$ the dispersive flux
given by:
$$\tilde{\boldsymbol{q}}_d = -\boldsymbol{D}\nabla C \qquad (2)$$
with $\boldsymbol{D}$, the dispersion tensor, expressed by:
$$\boldsymbol{D} = D_m\boldsymbol{I} + \left(\alpha_L - \alpha_T\right)\boldsymbol{q} \otimes \boldsymbol{q} / |\boldsymbol{q}| + \alpha_T|\boldsymbol{q}|\boldsymbol{I} \tag{3}$$

in which $\alpha_L$ and $\alpha_T$ are the longitudinal and transverse dispersivities [L], $D_m$ is the pore
water diffusion coefficient [$L^2T^{-1}$] and $\boldsymbol{I}$ is the unit tensor.
The water content $\theta$ and the Darcy velocity $\boldsymbol{q}$ are linked by the fluid mass conservation
equation in variably saturated porous media:
$$\frac{\partial \theta}{\partial t} + \nabla . \boldsymbol{q} = 0 \tag{4}$$

Substituting Eq. (4) into Eq. (1) yields the following advection-dispersion equation:
$$\theta\frac{\partial C}{\partial t} + \nabla.\left(\tilde{\boldsymbol{q}}_a + \tilde{\boldsymbol{q}}_d\right) - C\nabla.\boldsymbol{q} = 0 \tag{5}$$

In this work, we consider that the velocity $\boldsymbol{q}$ is obtained by solving Richards' equation using
the hybrid-MFE method. For a two-dimensional domain with a triangular mesh, $\boldsymbol{q}$ is
approximated inside each triangle $E$ using the lowest-order Raviart-Thomas (RT0) vectorial
basis functions $\boldsymbol{w}_j^E$:
$$\boldsymbol{q} = \sum_{j=1}^{3} Q_j^E \boldsymbol{w}_j^E \tag{6}$$

where $Q_j^E$ is the water flux across the edge $E_j$ of $E$ (see Figure 1) and $\boldsymbol{w}_j^E = \dfrac{1}{2|E|}\begin{pmatrix} x - x_j^E \\ y - y_j^E \end{pmatrix}$
is the typical RT0 basis functions (Younes $et\ al.$, 1999) with $\left(x_j^E, y_j^E\right)$ the coordinates of the
node $j$ opposite to the edge $E_j$ of $E$ and $|E|$, the area of $E$.

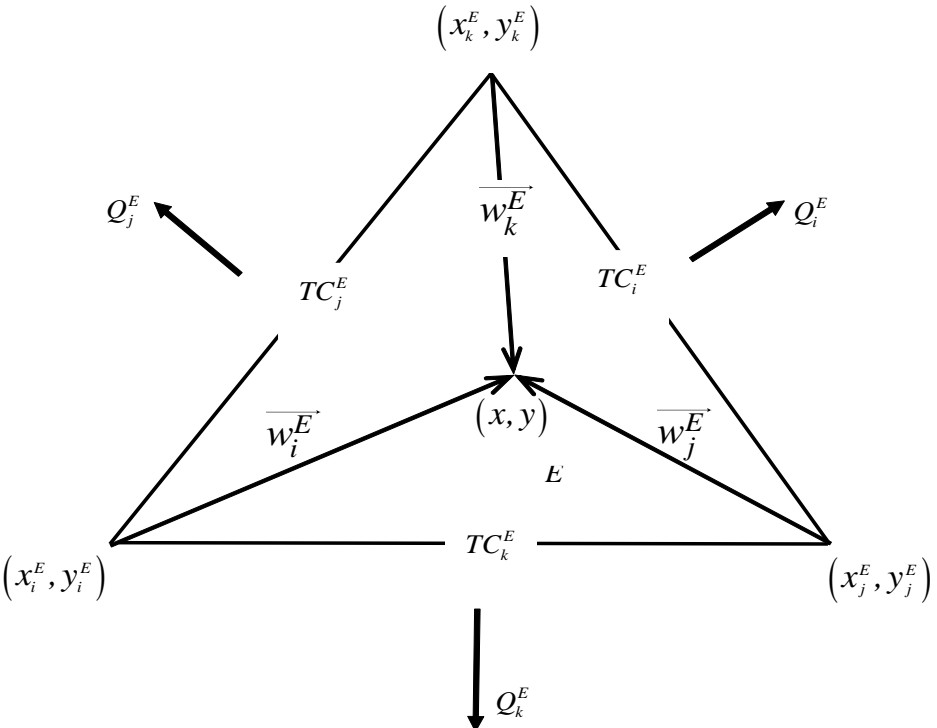


Figure 1: Vectorial basis functions for the MFE method.


To apply the hybrid-MFE method to the transport Eq. (5), we approximate the dispersive flux
$\tilde{\boldsymbol{q}}_d$ with RT0 vectorial basis functions as:

$$\tilde{\boldsymbol{q}}_d = \sum_{j=1}^{3} \tilde{Q}_j^{d,E} \boldsymbol{w}_j^E \tag{7}$$

where $\tilde{Q}_j^{d,E} = \int_{E_j} \tilde{\boldsymbol{q}}_d . \boldsymbol{\eta}_j^E$ is the dispersive flux across the edge $E_j$ of the element $E$ and $\boldsymbol{\eta}_j^E$ is
the outward unit normal vector to the edge $E_j$.
The variational formulation of Eq. (2) using the test function $\boldsymbol{w}_i^E$ yields:

$$\int_E \boldsymbol{D}^{-1} \tilde{\boldsymbol{q}}_d \boldsymbol{w}_i^E = \int_E C\nabla.\boldsymbol{w}_i^E - \sum_j \int_{E_j} C\boldsymbol{w}_i^E.\boldsymbol{\eta}_j^E \tag{8}$$

Substituting Eq. (7) into Eq. (8) and using properties of the basis functions $\boldsymbol{w}_j^E$ give
$$\sum_j \tilde{Q}_j^{d,E} \int_E \left( \boldsymbol{D}_E^{-1} \boldsymbol{w}_j^E \right) . \boldsymbol{w}_i^E = \frac{1}{|E|} \int_E C - \frac{1}{|E_i|} \int_{E_i} C \tag{9}$$
$$= C_E - TC_i^E$$

in which, $\boldsymbol{D}_E$ is the local dispersion tensor at the element $E$, $C_E$ is the mean concentration at
$E$ and $TC_i^E$ is the edge (trace) concentration (Lagrange multiplier) at the edge $E_i$.
Denoting the local matrix $\tilde{B}_{i,j}^E = \int_E \left( \boldsymbol{D}_E^{-1} \boldsymbol{w}_j^E \right) . \boldsymbol{w}_i^E$, the inversion of the system of Eq. (9) gives
the expression for the dispersive flux $\tilde{Q}_i^{d,E}$:
$$\tilde{Q}_i^{d,E} = \sum_j \tilde{B}_{i,j}^{E,-1} \left( C_E - TC_j^E \right) \tag{10}$$

Besides, the integration of the mass conservation Eq. (6) over the element $E$ writes
$$\int_E \theta \frac{\partial C}{\partial t} + \int_E \nabla . \tilde{\boldsymbol{q}}_a + \int_E \nabla . \tilde{\boldsymbol{q}}_d - \int_E C \nabla . \boldsymbol{q} = 0 \tag{11}$$

which becomes, using Green's formula,
$$\theta_E |E| \frac{\partial C_E}{\partial t} + \sum_i \int_{E_i} C \boldsymbol{q} . \boldsymbol{\eta}_i^E + \sum_i \int_{E_i} \tilde{\boldsymbol{q}}_d . \boldsymbol{\eta}_i^E - \int_E C \nabla . \boldsymbol{q} = 0 \tag{12}$$

where $\theta_E$ is the water content of the element $E$.
Substituting Eq. (2) into Eq. (12) yields
$$\theta_E |E| \frac{\partial C_E}{\partial t} + \sum_i \underbrace{\left( \tilde{Q}_i^{a,E} + \tilde{Q}_i^{d,E} \right)}_{\tilde{Q}_i^{t,E}} - C_E \sum_i Q_i^E = 0 \tag{13}$$

in which $\tilde{Q}_i^{t,E} = \tilde{Q}_i^{a,E} + \tilde{Q}_i^{d,E}$ is the total flux at the edge $E_i$ with $\tilde{Q}_i^{a,E}$ the advective flux given
by $\tilde{Q}_i^{a,E} = Q_i^E TC_i^E$ and $\tilde{Q}_i^{d,E}$ the dispersive flux given by Eq. (10).
The hybridization of the MFE method is performed in the following two steps:
1) The flux Eq. (10) is substituted into the mass conservation Eq. (13), which is then
discretized in time using the first-order implicit Euler scheme
$$\theta_E \frac{|E|}{\Delta t}\left(C_E^{\,n+1} - C_E^{\,n}\right) + \sum_i Q_i^E TC_i^{E,n+1} - C_E^{\,n+1} \sum_i Q_i^E + \tilde{\alpha}^E C_E^{\,n+1} - \sum_i \tilde{\alpha}_i^E TC_i^{E,n+1} = 0 \qquad (14)$$
in which $\tilde{\alpha}_i^E = \sum_j \tilde{B}_{i,j}^{E,-1}$ and $\tilde{\alpha}^E = \sum_i \tilde{\alpha}_i^E$.
Hence, the mean concentration at the new time level $C_E^{\,n+1}$ can be expressed as a function
of $TC_i^{E,n+1}$, the concentration at the edges of $E$, as follows:
$$C_E^{\,n+1} = \frac{1}{\beta_E}\sum_i\left(\tilde{\alpha}_i^E - Q_i^E\right)TC_i^{E,n+1} + \frac{\lambda_E}{\beta_E}C_E^{\,n} \qquad (15)$$
in which $\lambda_E = \theta_E \dfrac{|E|}{\Delta t}$ and $\beta_E = \left(\lambda_E + \tilde{\alpha}^E - \sum_i Q_i^E\right)$.
The mean concentration given by Eq. (15) is then substituted into the flux Eq. (10), which
allows expressing the dispersive flux $\tilde{Q}_i^{d,E,n+1}$ (the subscript n+1 will be omitted to alleviate
the notations) as only a function of the traces of concentration at edges $TC_i^{E,n+1}$:
$$\tilde{Q}_i^{d,E} = \sum_j\left(\frac{\tilde{\alpha}_i^E}{\beta_E}\left(\tilde{\alpha}_j^E - Q_j^E\right) - \tilde{B}_{i,j}^{E,-1}\right)TC_j^{E,n+1} + \frac{\lambda_E}{\beta_E}\tilde{\alpha}_i^E C_E^{\,n} \qquad (16)$$
2) The system to be solved is obtained by imposing the continuity of the total flux
$\left(\tilde{Q}_i^{t,E} + \tilde{Q}_i^{t,E'} = 0\right)$ as well as the continuity of the trace of concentration $\left(TC_i^{E,n+1} = TC_i^{E',n+1}\right)$
at the edge $i$ between the two elements $E$ and $E'$ (Figure 2).

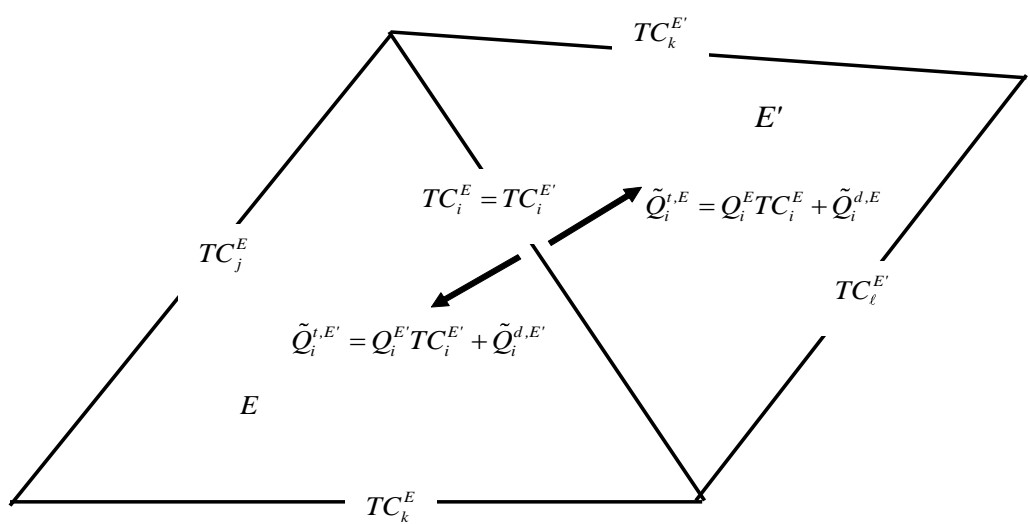


Figure 2: Continuity of concentration and total flux between adjacent elements with the
hybrid-MFE method.
Note that the advective flux $\tilde{Q}_i^{a,E}$ is continuous between $E$ and $E'$ because of the continuity
of the water flux and the continuity of the trace of concentration at the interface. Thus, for
the continuity of the total flux $\left(\tilde{Q}_i^{t,E} + \tilde{Q}_i^{t,E'} = 0\right)$, it is required that the dispersive flux is
continuous:
$$\tilde{Q}_i^{t,E} + \tilde{Q}_i^{t,E'} = \left(Q_i^E + Q_i^{E'}\right)TC_i^{E,n+1} + \tilde{Q}_i^{d,E} + \tilde{Q}_i^{d,E'} = \tilde{Q}_i^{d,E} + \tilde{Q}_i^{d,E'} = 0 \tag{17}$$

Using Eq. (16), we obtain:
$$\sum_j \left( \tilde{B}_{i,j}^{E,-1} - \frac{\tilde{\alpha}_i^E}{\beta_E}\left(\tilde{\alpha}_j^E - Q_j^E\right) \right) TC_j^{E,n+1} + \sum_j \left( \tilde{B}_{i,j}^{E',-1} - \frac{\tilde{\alpha}_i^{E'}}{\beta_{E'}}\left(\tilde{\alpha}_j^{E'} - Q_j^{E'}\right) \right) TC_j^{E',n+1}$$
$$= \frac{\lambda_E}{\beta_E}\tilde{\alpha}_i^E {C_E}^n + \frac{\lambda_{E'}}{\beta_{E'}}\tilde{\alpha}_i^{E'} {C_{E'}}^n \tag{18}$$

The continuity Eq. (18) is written for all mesh edges, and the resulting equations form the
final system to be solved for the traces of concentration at edges $TC_i^{E,n+1}$ as unknowns.
Note that the hybrid-MFE Eqs (18), obtained by approximating the dispersive flux with RT0
basis functions, is equivalent to the new MFE method proposed in Radu *et al.* (2011).
## 3.  The upwind and lumped MFE approaches
In this section, we recall the main principles of two existing approaches, developed to
improve the stability of the MFE solution of the transport equation. The first approach is the
upwind-hybrid MFE scheme of Radu et al. (2011), developed for advection dominated
transport. The second approach is the lumped hybrid-MFE method of Younes *et al.* (2006),
developed for dispersive transport.
**3.1 The upwind-hybrid MFE of Radu et al. (2011)**

In the case of advection-dominated transport, solving the hybrid-MFE Eq. (18) can yield
solutions with strong instabilities. A common way to avoid such instabilities is to use an
upwind scheme for the advective flux. Thus, for an element $E$, the advective flux
$\tilde{Q}_i^{a,E} = Q_i^E TC_i^E$ at the edge $i$ (common with the element $E'$), has to be calculated using either
the concentration from $E$ (if $Q_i^E > 0$) or the concentration from $E'$ (if $Q_i^E < 0$). Radu $et$ $al.$
(2011) suggested replacing the advective flux $\tilde{Q}_i^{a,E} = Q_i^E TC_i^E$ at the interface by:
$$\tilde{Q}_i^{a,E} = \begin{cases} Q_i^E C^E & if & Q_i^E > 0 \\ Q_i^E C^{E'} & if & Q_i^E < 0 \end{cases} \tag{19}$$

The advective term is now calculated using the upwind mean concentration, which can be that
of the element $E$ or of its adjacent element $E'$.
The advective flux of Eq. (19) is rewritten in the following condensed form
$$\tilde{Q}_i^{a,E} = Q_i^E \left( \tau_i^E C^E + \left( 1 - \tau_i^E \right) C^{E'} \right) \tag{20}$$

with $\tau_i^E = 1$ for an outflow $\left( Q_i^E > 0 \right)$ and $\tau_i^E = 0$ for an inflow $\left( Q_i^E < 0 \right)$.
However, this expression is incompatible with the hybridization procedure. Indeed, if we
replace, in the Eq. (14), the advective term $Q_i^E TC_i^E$ by Eq. (20), the latter will contain both
$C^E$ and $C^{E'}$. Thus, the first step of the hybridization procedure cannot allow expressing
$C_E^{n+1}$ as only a function of $TC_i^{E,n+1}$ as in the Eq. (15).
To avoid this difficulty, Radu $et$ $al.$ (2011) suggested replacing, $C^{E'}$ by the following
expression:
$$C^{E'} \simeq 2TC_i^E - C^E \tag{21}$$

This approximation is based on the assumption that $TC_i^E \simeq \left( C^E + C^{E'} \right) / 2$.
Plugging Eq. (21) into Eq. (20), the advective flux $\tilde{Q}_i^{a,E}$ depends only on the variables of the
element $E$ (mean concentration $C^E$ and edge concentration $TC_i^E$):

$$\tilde{Q}_i^{a,E} = Q_i^E \left( \tau_i^E C^E - \left(1 - \tau_i^E\right) C^E + 2\left(1 - \tau_i^E\right) TC_i^E \right) \tag{22}$$

Eq. (22) can then be used to replace the advective term $Q_i^E TC_i^{E,n+1}$ in Eq. (14), and thus the

hybridization procedure allows to express $C_E^{n+1}$ as a function of $TC_i^{E,n+1}$ as in the Eq. (15).

Then, the expression of $C_E^{n+1}$ is substituted into the dispersive flux Eq. (10), and the final

system is obtained by prescribing continuity of the total flux $\left( \tilde{Q}_i^{t,E} + \tilde{Q}_i^{t,E'} = 0 \right)$ at the interface

between $E$ and $E'$. This scheme was shown to be more efficient (by using a sparser system

matrix with fewer unknowns) than the non-hybrid upwind mixed method of Dawson (1978).

The two methods yielded optimal first order convergence in time and space (Brunner *et al.*,

2014).

The assumption given by Eq. (21) can be a rough approximation, especially in the case of a

heterogeneous domain where dispersion can vary with several orders of magnitudes from

element to element. For such a situation, the edge concentration can be significantly different

from the average of the mean concentrations of adjacent elements. Furthermore, the advective

flux is not uniquely defined at the interface and can be different for the two adjacent elements

$E$ and $E'$. For instance, in the case of $Q_i^E = Q > 0$, the advective flux leaving the element $E$ is

$\tilde{Q}_i^{a,E} = QC^E$, whereas the flux entering the element $E'$ is $\tilde{Q}_i^{a,E'} = Q\left(2TC_i^E - C^{E'}\right)$ which could

be different as $TC_i^E$ is not necessarily the mean of $C^E$ and $C^{E'}$. In this situation, because of

the discontinuity of the advective flux, the dispersive flux will not be continuous at the

interface since the continuity is prescribed only for the total flux.

**3.2 The lumped hybrid-MFE scheme for dispersion transport**

In this section, we recall the main principles of the lumped hybrid-MFE method of Younes *et*

*al.* (2006), developed to improve the stability of the MFE solution in the case of dispersive

transport.

Considering only dispersion, Eq. (5) simplifies to:

$$\theta \frac{\partial C}{\partial t} + \nabla . \tilde{\boldsymbol{q}}_d = 0 \tag{23}$$

As detailed above, the hybrid MFE method for Eq. (23) is based on two stages:
• *Stage1: discretization of the transient mass conservation equation over the element E:*
The integration of the mass conservation Eq. (23) over the element $E$ gives (see Eq.

13):

$$\theta_E |E| \frac{\partial C_E}{\partial t} + \sum_i \tilde{Q}_i^{d,E} = 0 \tag{24}$$

• *Stage2: imposing the continuity of the flux across the edge i sharing the two elements*
*E and E':*

$$\tilde{Q}_i^{d,E} + \tilde{Q}_i^{d,E'} = 0 \tag{25}$$

Note that the continuity equation (25) can be interpreted as a steady state mass conservation
equation at the edge level. Hence, the hybrid MFE discretization uses the transient mass
conservation equation at the element level, given by Eq. (24), and the steady state mass
conservation at the edge level, given by Eq. (25). With the lumped hybrid MFE method of
Younes *et al.* (2006), the transient term is taken into account at the edge level. Hence, the
lumped formulation uses a steady state mass conservation equation at the element level and a
transient mass conservation equation at the edge level. The two stages of the lumped hybrid
MFE are as follows:
• *Stage1: discretization of the steady-state mass conservation equation over E:*
The steady-state transport over the element $E$ writes:

$$\sum_i \tilde{\underline{Q}}_i^{d,E} = 0 \tag{26}$$

where $\tilde{\underline{Q}}_i^{d,E}$ is the steady-state dispersive flux across the edge $E_i$.
Therefore, the mean concentration of Eq. (15) becomes
$$C_E = \sum_i \frac{\tilde{\alpha}_i^E}{\tilde{\alpha}^E} TC_i^E \tag{27}$$

and using Eq. (16), the steady-state dispersive flux writes
$$\underline{\tilde{Q}}_i^{d,E} = \sum_j \left( \frac{\tilde{\alpha}_i^E \tilde{\alpha}_j^E}{\tilde{\alpha}^E} - \tilde{B}_{i,j}^{E,-1} \right) TC_j^E \tag{28}$$

• *Stage2: discretization of the transient mass conservation equation over the lumping*
*region* $R_i$
The edge centered finite volume discretization of the transient transport Eq. (23) over
the lumping region $R_i$ (hatched area in Figure 3), associated with the edge $i$, writes:
$$\int_{R_i} \theta \frac{\partial C}{\partial t} + \int_{R_i} \nabla . \tilde{q}_d = 0 \tag{29}$$

where the lumping regions $R_i$ is formed by the two simplex regions $S_i^E$ and $S_i^{E'}$, for
an inner edge $i$ sharing the two elements $E$ and $E'$, and by the sole simplex region
$S_i^E$ for a boundary edge. The simplex region $S_i^E$ is defined by joining the centre of $E$
with the nodes $j$ and $k$ forming the edge $i$.

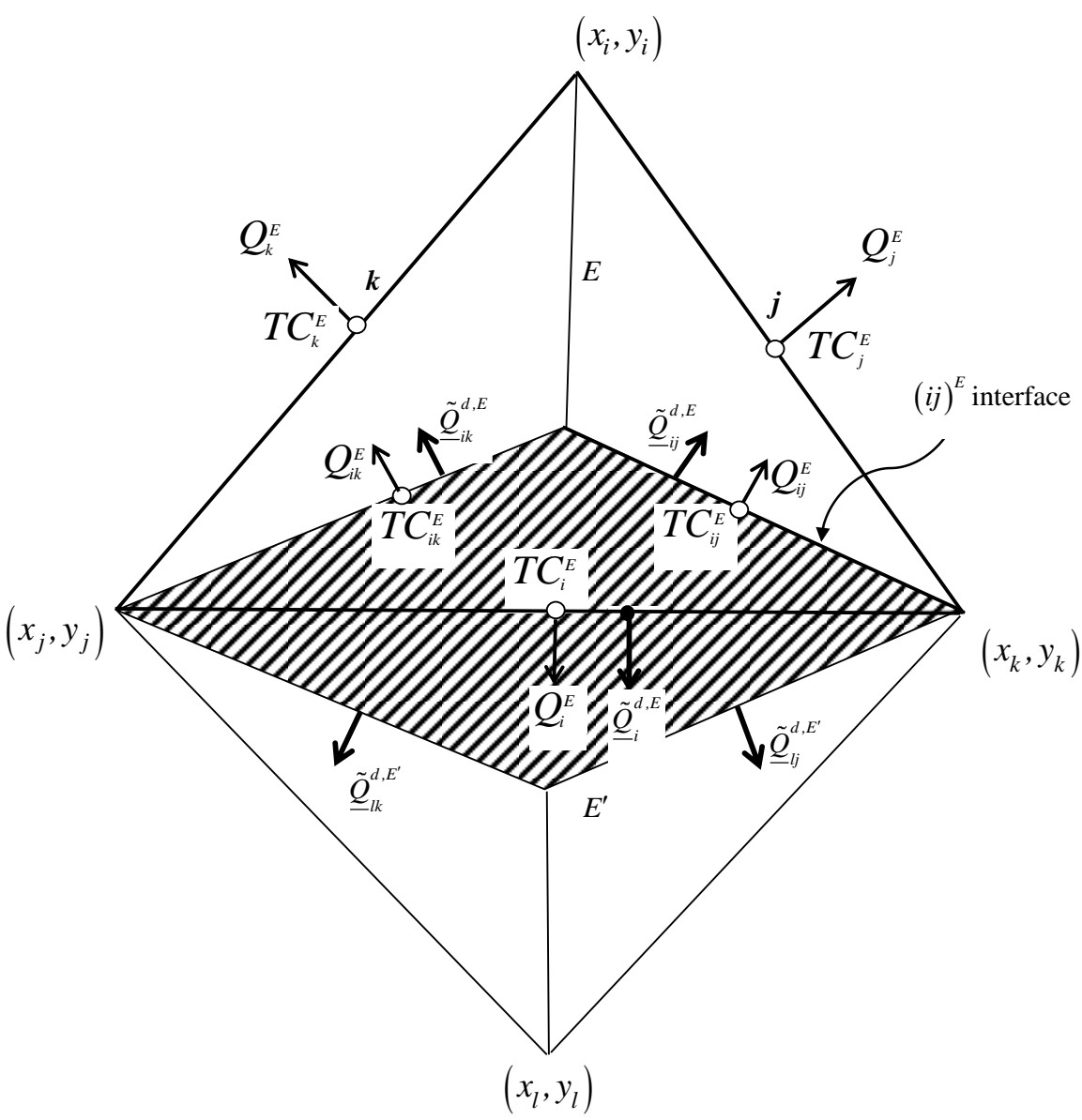


Figure 3: The lumping region $R_i$ associated with the edge $i$, sharing the elements $E$ and

$E'$ and formed by the two simplex regions $S_i^E$ and $S_i^{E'}$.

Associating the edge concentration $TC_i^E$ to $R_i$ (see Figure 3 for notations), Eq. (29) gives

$$\left\{ \frac{|E|}{3} \theta_E \frac{\partial TC_i^E}{\partial t} + \underline{\tilde{Q}}_{ij}^{d,E} + \underline{\tilde{Q}}_{ik}^{d,E} \right\} + \{\ \}' = 0 \tag{30}$$

in which $\underline{\tilde{Q}}_{ij}^{d,E}$ and $TC_{ij}^E$ are respectively the dispersive flux and the concentration at the

interior interface $(ij)^E$ between the simplex regions $S_i^E$ and $S_j^E$. The shortcut $\{\ \}'$

designates the same contribution as $\{\ \}$, but of the adjacent element $E'$, in the case of Eq.
(30), it corresponds to $\dfrac{|E'|}{3}\theta_{E'}\dfrac{\partial TC_i^E}{\partial t}+\underline{\tilde{Q}}_{ij}^{d,E'}+\underline{\tilde{Q}}_{ik}^{d,E'}$ .
Besides, applying the steady-state dispersive transport Eq. (26) on the simplex region $S_i^E$
yields:

$$\underline{\tilde{Q}}_{ij}^{d,E}+\underline{\tilde{Q}}_{ik}^{d,E}+\underline{\tilde{Q}}_{i}^{d,E}=0 \tag{31}$$

Finally, substituting Eq. (28) and Eq. (31) into the transport Eq. (30) give the final system
to solve with the lumped hybrid MFE scheme:

$$\left\{\dfrac{|E|}{3}\theta_E\dfrac{\partial TC_i^E}{\partial t}+\sum_j\left(\tilde{B}_{i,j}^{E,-1}-\dfrac{\tilde{\alpha}_i^E\tilde{\alpha}_j^E}{\tilde{\alpha}^E}\right)TC_j^E\right\}+\{\ \}'=0 \tag{32}$$

Note that
1.  The lumped hybrid formulation Eq. (32) and the standard hybrid formulation (Eqs (24)-

(25)) are exactly the same in the case of steady state diffusion transport.

2.  In the lumped formulation Eq (32), the term of mass (with time derivative) has a

contribution only on the diagonal term of the final system matrix. This improves the

monotonous character of the solution (see Younes *et al.*, 2006). For instance, in the case

of an acute triangulation, the maximum principle is respected by the lumped

formulation Eq. (32) whatever the heterogeneity of the porous medium (Younes *et al.*,

2006).

3.  Contrarily to the standard hybrid-MFE scheme, where the discretization of the temporal

derivative performed in Eq. (14) was necessary to obtain the final system given by Eq.

(18), the lumped scheme given by Eq. (32) keeps the time derivative continuous which

allows the use of efficient high order temporal discretization methods via the MOL.

4.  In the case of 2D triangular elements, the lumped formulation Eq. (32) is algebraically

equivalent to the nonconforming Crouzeix-Raviart (Crouzeix and Raviart, 1973) finite

element method (see Younes *et al.*, 2008). The nonconforming Crouzeix-Raviart
method uses the chapeau functions as basis functions to approximate the concentration,
like the standard finite element method, but seed nodes are the midpoints of the edges.

## 4. The new upwind-hybrid MFE scheme for advection-dispersion transport

To avoid the rough approximation (21), we develop hereafter a new upwind-MFE scheme
where the advection term is calculated using upwind edge concentration instead of upwind
mean concentration of the element *E*. The idea of the scheme is to extend the lumped hybrid-
MFE procedure to transport by both advection and dispersion and to use an upwind edge
centered FV scheme to avoid unphysical oscillations caused by the hyperbolic nature of
advection.
The integration of the whole mass conservation Eq. (5) over the lumping region $R_i$ writes:

$$\int_{R_i}\theta\frac{\partial C}{\partial t}+\int_{R_i}\nabla.\left(qC\right)+\int_{R_i}\nabla.\tilde{q}_d-\int_{R_i}C\nabla.q=0 \tag{33}$$

Using notations of Figure 3, we obtain

$$\left\{\frac{|E|}{3}\theta_E\frac{\partial TC_i^E}{\partial t}+Q_{ij}^E TC_{ij}^E+Q_{ik}^E TC_{ik}^E+\underline{\tilde{Q}}_{ij}^{d,E}+\underline{\tilde{Q}}_{ik}^{d,E}-TC_i^E\left(Q_{ij}^E+Q_{ik}^E\right)\right\}+\left\{\ \right\}'=0 \tag{34}$$

in which $Q_{ij}^E$ is the water flux at the interior interface $\left(ij\right)^E$, evaluated using the RT0
approximation of the velocity given by Eq. (6), which yields

$$Q_{ij}^E=\frac{1}{3}\left(Q_j^E-Q_i^E\right) \tag{35}$$

Using Eq (28) and Eq. (31) and denoting $\lambda_E=\theta_E\dfrac{|E|}{3}$, Eq. (34) becomes

$$\left\{\lambda_E\frac{\partial TC_i^E}{\partial t}+\sum_j\left(\tilde{B}_{i,j}^{E,-1}-\frac{\tilde{\alpha}_i^E\tilde{\alpha}_j^E}{\tilde{\alpha}^E}\right)TC_j^E+Q_{ij}^E TC_{ij}^E+Q_{ik}^E TC_{ik}^E-\left(Q_{ij}^E+Q_{ik}^E\right)TC_i^E\right\}+\left\{\ \right\}'=0 \tag{36}$$

The interior concentration $TC_{ij}^{E}$ at the interface between the simplex regions $S_i^E$ and $S_j^E$ is
calculated using an upwind scheme (See Figure 3) defined by:
$$TC_{ij}^{E} = \tau_{ij}^{E}TC_i^E + \left(1-\tau_{ij}^{E}\right)TC_j^E \tag{36}$$

with $\tau_{ij}^{E}=1$ if $\left(Q_{ij}^{E} \geq 0\right)$, else $\tau_{ij}^{E}=0$
Thus, the final system to solve becomes,
$$\left\{ \lambda_E \frac{\partial TC_i^E}{\partial t} + \sum_j \left( \tilde{B}_{i,j}^{E,-1} - \frac{\tilde{\alpha}_i^E \tilde{\alpha}_j^E}{\tilde{\alpha}^E} \right) TC_j^E + Q_{ij}^E \left(1-\tau_{ij}^E\right)\left(TC_j^E - TC_i^E\right) + Q_{ik}^E \left(1-\tau_{ik}^E\right)\left(TC_k^E - TC_i^E\right) \right\}$$

$$+ \{ \ \}' = 0$$

(37)

In the case of a first-order Euler implicit time discretization, Eq. (37) becomes
$$\left\{ \begin{array}{l} \sum_j \left( \tilde{B}_{i,j}^{E,-1} - \frac{\tilde{\alpha}_i^E \tilde{\alpha}_j^E}{\tilde{\alpha}^E} \right) TC_j^{E,n+1} + \lambda_E TC_i^{E,n+1} + Q_{ij}^E \left(1-\tau_{ij}^E\right)\left(TC_j^{E,n+1} - TC_i^{E,n+1}\right) \\ + Q_{ik}^E \left(1-\tau_{ik}^E\right)\left(TC_k^{E,n+1} - TC_i^{E,n+1}\right) - \lambda_E TC_i^{E,n} \end{array} \right\} + \{ \ \}' = 0 \quad (38)$$

where $\lambda_E = \theta_E \dfrac{|E|}{3\Delta t}$.
It is easy to see that, due to upwinding, the system matrix corresponding to Eq. (38) is always
an *M*-matrix (a non singular matrix with $m_{ii}>0$, $m_{ij}\leq 0$) in the case of transport by advection.
The *M*-matrix property insures the stability of the scheme since it guaranties the respect of the
discrete maximum principle *i.e.* local maxima or minima will not appear in the $C$ solution in
a domain without local sources or sinks.
Further, Eq. (37) expresses the total exchange between $E$ and $E$' and therefore reflects the
continuity of the total advection-dispersion flux between them. Both advective and dispersive
fluxes are continuous between the adjacent elements $E$ and $E$'. The advective flux, calculated
using the upwind edge concentration, is uniquely defined at the interface of the lumping
region and is therefore continuous. As a consequence, the dispersive flux is also continuous
between *E* and *E*' since the total flux is continuous at the interface between them.

## 5. Numerical Experiments

In this section, a first test case dealing with transport in saturated porous media is simulated
with the standard hybrid-MFE and the new upwind-MFE schemes. The results are compared
against an analytical solution in order to validate the new developed scheme and to show its
robustness for solving advection-dominated transport problems compared to the standard one.
The second test case deals with transport in the unsaturated zone and aims to investigate the
robustness of the new scheme when combined with the MOL for solving highly nonlinear
problems.

**5.1 Transport in saturated porous media: comparison against a 2D analytical solution**

The hybrid and upwind MFE formulations are compared against the analytical solution
developed by Leij and Dane (1990) for a simplified 2D transport problem (Figure 4). The test
case has been employed by Putti *et al.* (1990) and Siegel *et al.* (1997) for the verification of
transport codes. It deals with the contamination from the left boundary of a 2D rectangular
domain of dimension $(0,100) \times (0,40)$.

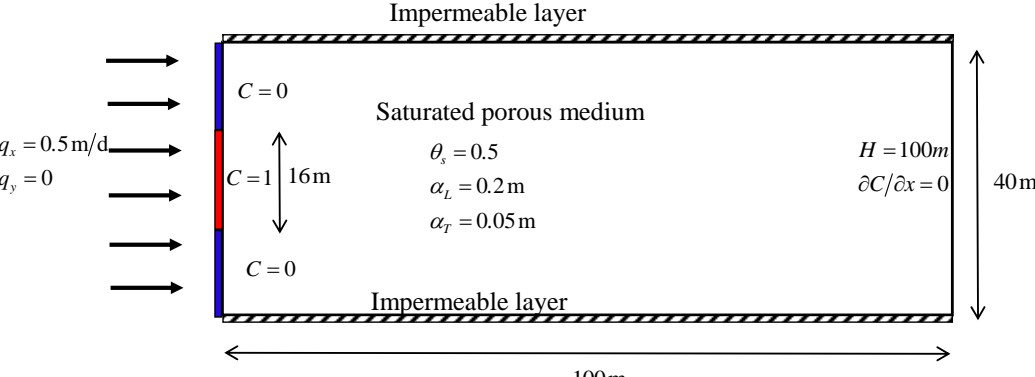

Figure 4: Description of the problem of the contamination of a 2D saturated porous medium.
The boundary conditions for the transport are of Dirichlet type at the inflow (left vertical
boundary), with

$$C = \begin{cases} 0 & \text{for } x = 0 \text{ and } 0 \le y < 12 \\ 1 & \text{for } x = 0 \text{ and } 12 \le y \le 28 \\ 0 & \text{for } x = 0 \text{ and } 28 < y \le 40 \end{cases} \quad (39)$$


A zero diffusive flux is imposed at the right vertical outflow boundary. The top and bottom
are impermeable boundaries. A uniform horizontal flow occurs from left to right with a
constant flux $q_x = 0.5$ m/day prescribed at the left vertical boundary and a fixed head
$H = 100\,\text{m}$ at the right vertical boundary. The longitudinal and transverse dispersivities are
$\alpha_L = 0.2m$ and $\alpha_T = 0.05m$, respectively. The domain is discretized with a fine unstructured
triangular mesh formed by 33216 elements, and the simulation is performed for a final
simulation time T = 30 days using the Euler-implicit time discretization with a fixed time step
of 0.1 day. The linear systems are solved in each time step with a direct solver using an
unsymmetric-pattern multifrontal method and a direct sparse LU factorization (UMFPACK).
The analytical solution of this test case for an infinite domain is given by Leij and Dane

(1990):

$$C_{analy}(x,y,t) = \frac{x}{(16\pi\alpha_L)^{1/2}} \int_0^T \tau^{-3/2} \left\{ erf\left[ \frac{y-12}{(4\alpha_T\tau)^{1/2}} \right] + erf\left[ \frac{28-y}{(4\alpha_T\tau)^{1/2}} \right] \right\} exp\left[ -\frac{(x-\tau)^2}{4\alpha_L\tau} \right] d\tau \quad (40)$$
with $erf(x) = \frac{2}{\sqrt{\pi}} \int_0^x exp(-\tau^2) d\tau$.

The final distributions of the concentration with both hybrid-MFE and upwind-MFE schemes
are depicted in Figure 5. Although we have used an unstructured mesh, the two schemes yield
almost symmetrical results. The hybrid-MFE scheme (Figure 5a) yields a solution with
unphysical oscillations. Indeed, around 1.2 % of the contaminated region (*i.e.* the region with
$|C| \ge 10^{-5}$) exhibits unphysical oscillations with 0.4 % of the contaminated region with
$C \le -10^{-3}$ and 0.8 % of the contaminated region with $C \ge 1.001$. These unphysical
oscillations, although they seem moderate, can be dramatic, for instance, when dealing with
reactive transport where some reactions occur only if the concentration excesses a certain
threshold. The solution obtained with the new upwind formulation (Figure 5b) is monotone
(all concentrations are between 0 and 1) which is in agreement with the physics. However,
these results come at the expense of some numerical diffusion added to the solution. To
appreciate the quality of both solutions and validate the upwind-MFE method, we compare
the concentration profile of the two methods to the analytical solution of Leij and Dane (1990)
for a horizontal section located at y = 20 m and a vertical section located at x = 20 m.

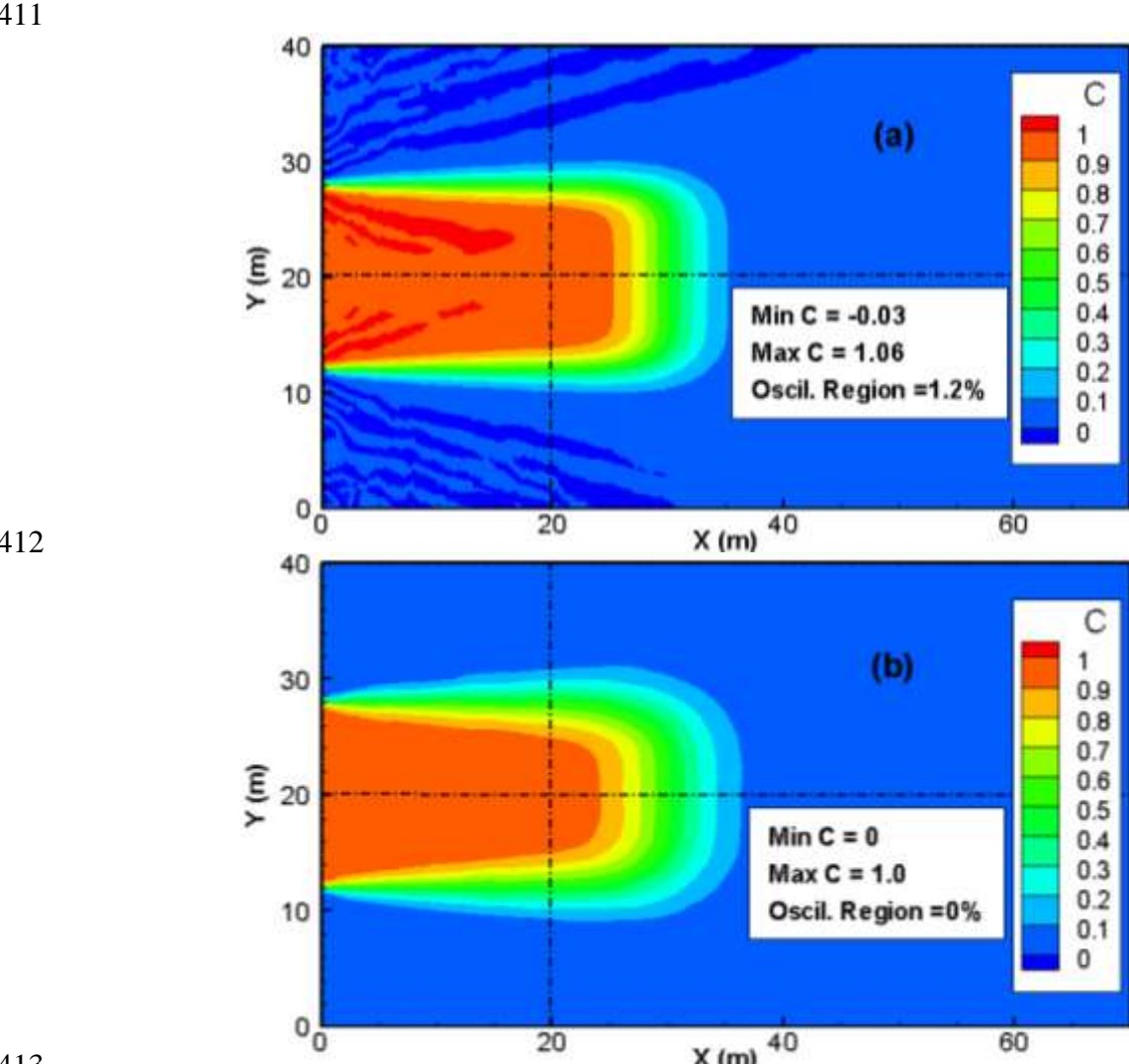


Figure 5: Concentration distribution with the hybrid-MFE and the upwind-MFE methods for

the 2D saturated transport problem (only the region $x \leq 70$ m is depicted).


The results of figure 6 show that the solution of both hybrid-MFE and upwind-MFE methods
are in very good agreement with the analytical solution, which validates the new upwind-
MFE numerical model. Note, however, that a small numerical diffusion is observed with the
upwind-MFE solution, which is especially visible in figure 6b. Indeed, for the simulated
problem, the transverse dispersivity is much smaller than the longitudinal one, and, as a
consequence, the concentration front is sharper in the vertical section than in the horizontal
one. This explains why the numerical diffusion generated by the upwind-MFE method is
more pronounced in Figure 6b than in Figure 6a.

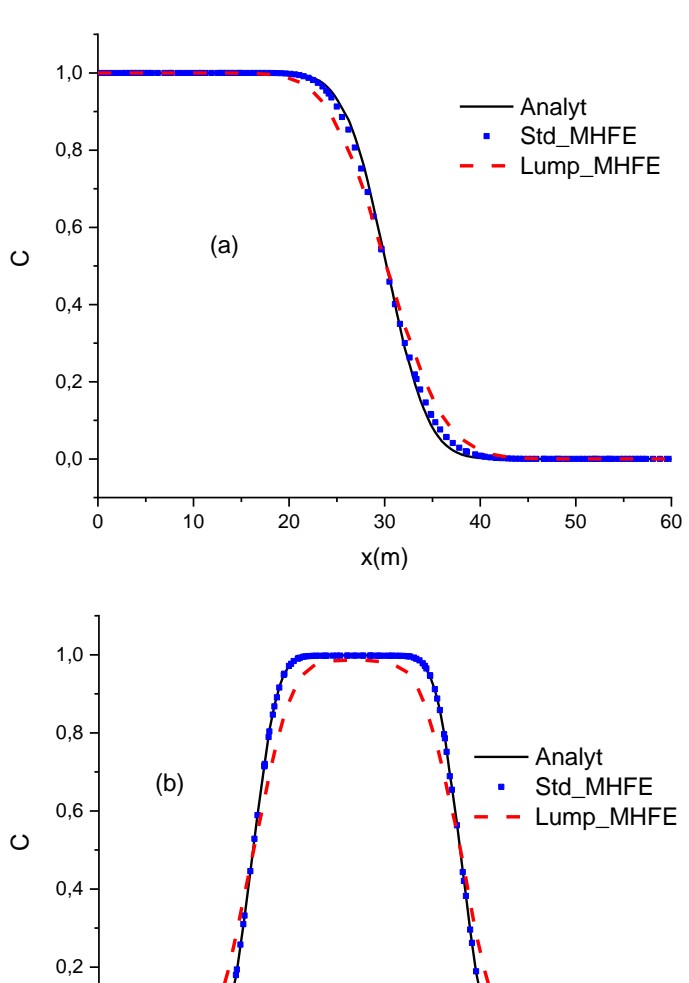



Figure 6: Concentration profiles at y = 20m (a) and x = 20 m (b) with the analytical, hybrid-

MFE and upwind-MFE solutions.

The test problem is then simulated using different mesh refinements to investigate the order of
convergence of the new method. We start with a uniform mesh formed by 1000 triangles and
a time step $\Delta t = 0.1s$. In each level of refinement, each triangle is subdivided into four similar
triangles, by joining the three mid-edges and the time step $\Delta t$ is halved. The following error
is computed (Brunner *et al.*, 2014):
$$Er = \left\{ \left\| C_{analyt}\left(t^N\right) - C\left(t^N\right) \right\|_0^2 + \Delta t \sum_{n=1}^{N} \left\| \tilde{q}_{analyt}^t\left(t^n\right) - \tilde{q}^t\left(t^n\right) \right\|_0^2 \right\}^{1/2} \tag{39}$$

where $\tilde{q}^t = \tilde{q}_a + \tilde{q}_d$ is the total advection-dispersion flux and $N$ the total number of time
steps.
The runs are performed on a single computer with an Intel Xeon E-2246G processor and 32
GB memory. The results of the computations, summarized in Table 1, clearly show optimal
first order convergence in space and time for the developed upwind-hybrid MFE method.

| Ref. level | # unknowns | Error $Er$ | Reduction | CPU time (s) |
|:---:|:---:|:---:|:---:|:---:|
| 1 | 1535 | 2.55 | | 4.9 |
| 2 | 6070 | 1.296 | 1.97 | 38.6 |
| 3 | 24140 | 0.655 | 1.98 | 272 |
| 4 | 96280 | 0.329 | 1.99 | 2068 |
| 5 | 384560 | 0.165 | 2.00 | 16567 |

Table 1: Numerical results for the new upwind-hybrid MFE method.

**5.2 Transport in a variably-saturated porous medium**
In this test case, the developed upwind-MFE method is combined with the MOL for solving
contaminant transport in a variably-saturated porous medium. The advection-dispersion
equation is transformed to an Ordinary Differential Equation (ODE) using the new upwind-
MFE formulation for the spatial discretization, whereas the time derivative is maintained
continuous. Therefore, high-order time integration methods included in efficient ODE solvers
can be employed. With these solvers, both the time step size and the order of the time
integration can vary during the simulation to deliver accurate results in an acceptable
computational time.
To investigate the robustness and efficiency of the combination of the developed upwind-
MFE method with the MOL, we simulate in this section the problem of contaminant
infiltration into a variably-saturated porous medium.

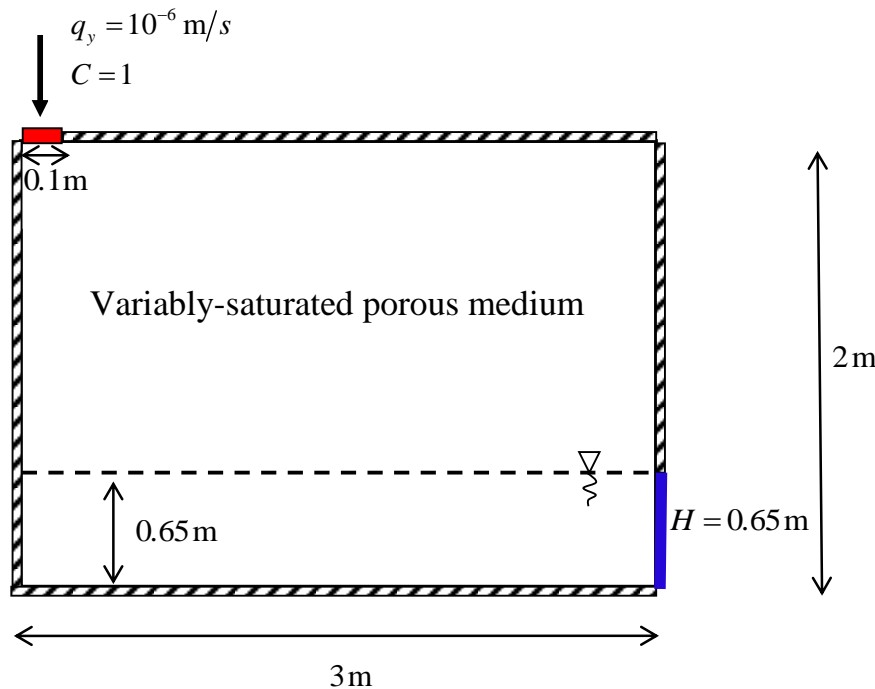


Figure 7: Description of the problem of contaminant infiltration into a 2D variably-saturated
porous medium.

The domain (Figure 7) is a rectangular box of 3m × 2m, filled with sand, with an initial water
table at 0.65m and hydrostatic pressure distribution. An infiltration of a tracer contaminant is
applied over the left-most 0.1m of the surface with a constant flux of $10^{-6}\,\mathrm{m/s}$. The right
vertical side has a fixed head $H = 0.65\,\mathrm{m}$ below the water table and an impermeable boundary
above it. The left vertical side as well as the upper (except the infiltration zone) and bottom
boundaries are impermeable boundaries.
In this problem, the flow and transport are coupled by the velocity, which is obtained by
solving the following pressure-head form of the nonlinear Richards' equation:
$$\left( c(h) + S_S \frac{\theta}{\theta_S} \right) \frac{\partial H}{\partial t} + \nabla \cdot \boldsymbol{q} = 0 \tag{40}$$

$$\boldsymbol{q} = -k_r \boldsymbol{K} \nabla H \tag{41}$$

with $S_S$ the specific mass storativity related to head changes [L$^{-1}$], $H = h + y$ the equivalent
head [L], $h = \dfrac{P}{\rho g}$ the pressure head, $P$ the pressure [Pa], $\rho$ the fluid density [ML$^{-3}$], $g$ the
gravity acceleration [LT$^{-2}$], $y$ the upward vertical coordinate [L], $c(h)$ the specific moisture
capacity [L$^{-1}$], $\theta_S$ the saturated water content [L$^3$L$^{-3}$], $\boldsymbol{q}$ the Darcy velocity [LT$^{-1}$],
$\boldsymbol{K} = \dfrac{\rho g}{\mu} \boldsymbol{k}$ the hydraulic conductivity [LT$^{-1}$], $\boldsymbol{k}$ the permeability [L$^2$], $\mu$ the fluid dynamic
viscosity [ML$^{-1}$T$^{-1}$] and $k_r$ the relative conductivity [-].
We use the standard van Genuchten (1980) model for the relationship between water content
and pressure head:
$$S_e = \frac{\theta(h) - \theta_r}{\theta_s - \theta_r} = \begin{cases} \dfrac{1}{\left(1 + |\alpha h|^n\right)^m} & h < 0 \\ 1 & h \geq 0 \end{cases} \tag{42}$$

where $\alpha$ [L$^{-1}$] and $n$ [-] are the van Genuchten parameters, $m = 1 - 1/n$, $S_e$ [-] is the effective
saturation and $\theta_r$ [-] is the residual water content. The conductivity-saturation relationship is
derived from the Mualem (1976) model:
$$k_r = S_e^{1/2} \left[ 1 - \left( 1 - S_e^{1/m} \right)^m \right]^2 \tag{43}$$

The material properties of the test problem are given in Table 2.

| Parameters | |
| --- | --- |
| $\theta_r$ | 0.01 |
| $\theta_s$ | 0.3 |
| $\alpha$ (cm$^{-1}$) | 0.033 |
| $n$ | 4.1 |
| $K$ (cm/s) | $10^{-2}$ |
| $S_s$ (cm$^{-1}$) | $10^{-10}$ |
| $D_m$ (m$^2$/s) | $10^{-9}$ |
| $\rho$ (kg/m$^3$) | 1000 |
| $\mu$ (kg/m/s) | 0.001 |


Table 2: Parameters for the problem of infiltration into a 2D variably-saturated porous
medium.

The simulation is performed for 80 hours using a triangular mesh formed by 4273 triangular
elements. Two test cases are investigated. In the first test case, the longitudinal and transverse
dispersivities are $\alpha_L = 0.03m$ and $\alpha_T = 0.003m$, respectively. The second test case is less
diffusive with $\alpha_L = 0.01m$ and $\alpha_T = 0.001m$.
The coupled nonlinear flow-transport system is solved using the MOL, which allows the use
of efficient high-order time integration methods, for both the hybrid-MFE and the upwind-
MFE schemes. To this aim, a hybrid-MFE formulation with continuous time derivative was
developed by extending the lumping procedure, developed in Younes *et al*. (2006) for the
flow equation, to the advection-dispersion transport Eq. (5).
The time integration is performed with the DASPK time solver which uses an efficient
automatic time-stepping scheme based on the Fixed Leading Coefficient Backward
Difference Formulas (FLCBDF). The linear systems arising at each time step are solved with
the preconditioned Krylov iterative method. The nonlinear problem is linearized using the
Newton method with a numerical approximation of the Jacobian matrix.
The results of the hybrid-MFE and the upwind-MFE methods are depicted in Figure 8 for the
first test case involving high dispersion. Good agreement can be observed between the results
of the hybrid-MFE (Figure 8a) and upwind-MFE (Figure 8b) schemes when combined with
the MOL. In these figures, the contaminant progresses essentially vertically through the
unsaturated zone of the soil. When the saturated zone is reached, the contaminant progresses
horizontally and remains close to the water table. Note that the results of both schemes are
stable and free from unphysical oscillations (Figures 8a and 8b).

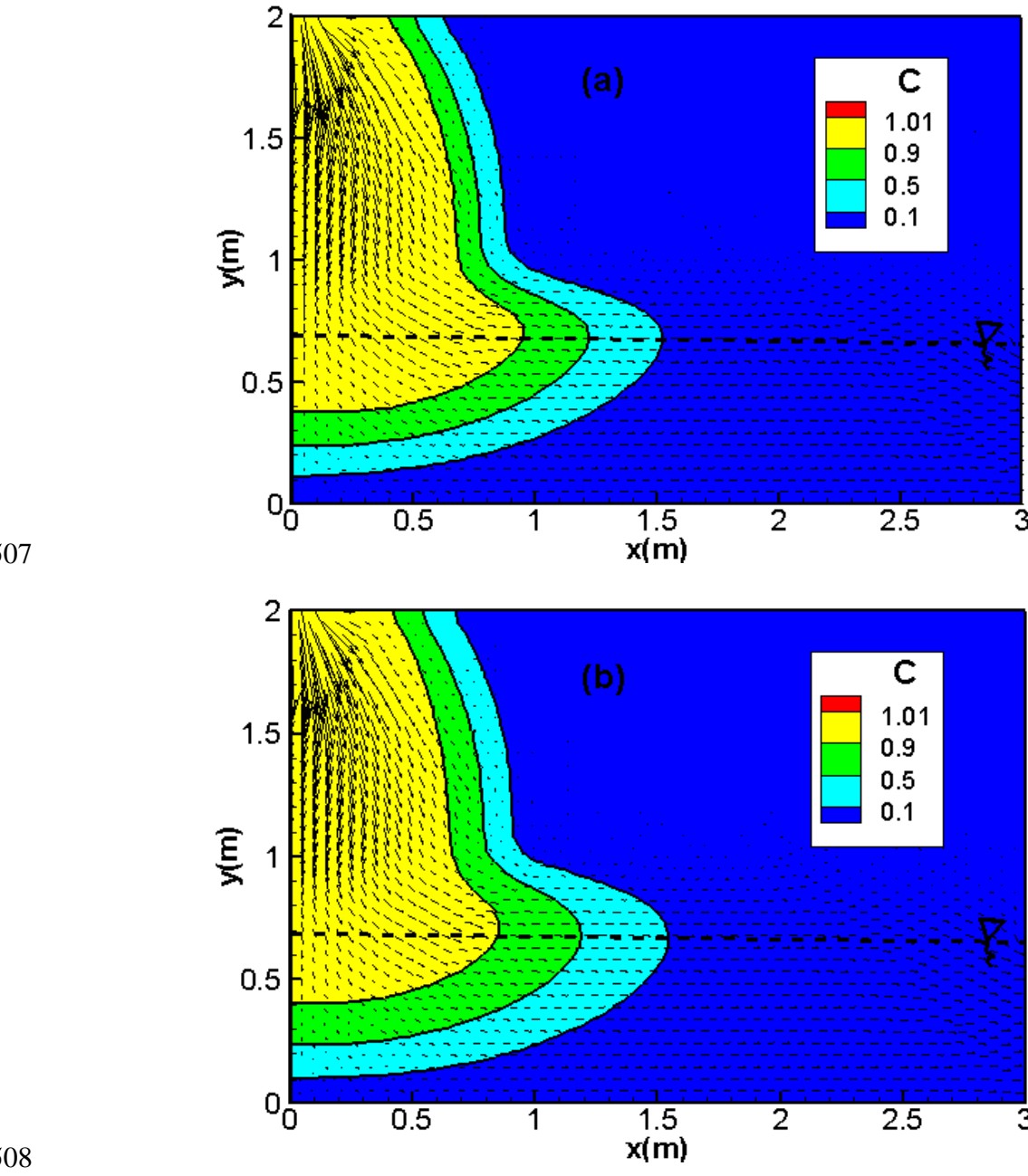


Figure 8: Concentration distribution, with the hybrid-MFE (a) and the upwind-MFE (b)

schemes for the transport problem with high dispersion in a variably-saturated porous

medium.

For the second test case with lower dispersion $\left( \alpha_L = 0.01m,\ \alpha_T = 0.001m \right)$, the hybrid-MFE
method yields unstable results containing unphysical oscillations (red color in Figure 9a).
These oscillations hamper the convergence of the numerical model, and severe convergence
issues can be encountered if we further decrease the dispersivity values. The results of the
upwind-MFE scheme are monotone and do not contain any unphysical oscillation (Figure 9b).
These results point out the robustness of the new upwind MFE method for transport in
saturated and unsaturated porous media. The developed transport scheme has recently been
successfully combined with the MFE method for fluid flow to simulate nonlinear flow and
transport in unsaturated fractured porous media using the 1D-2D discrete fracture matrix
(DFM) approach (Younes et al., 2022b).

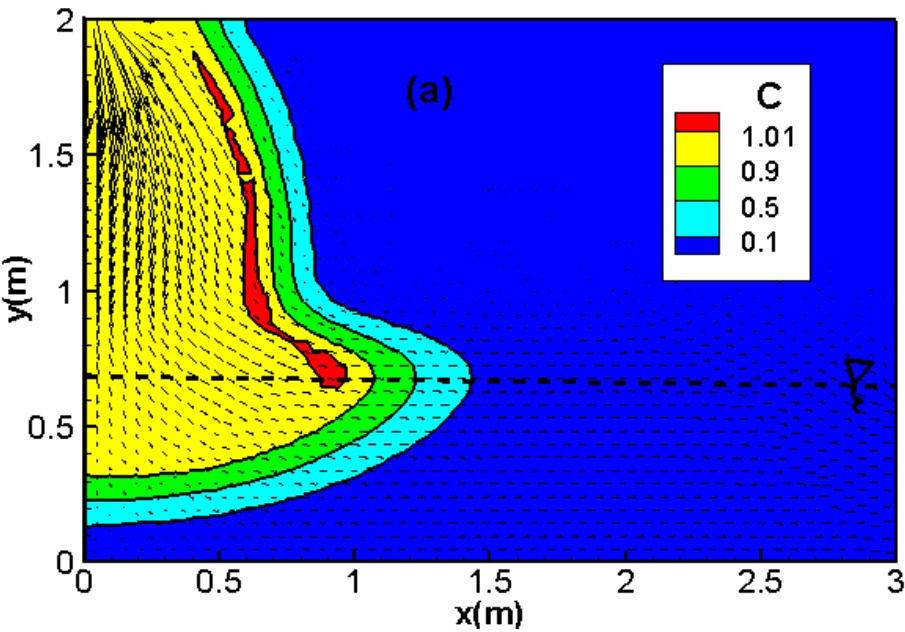


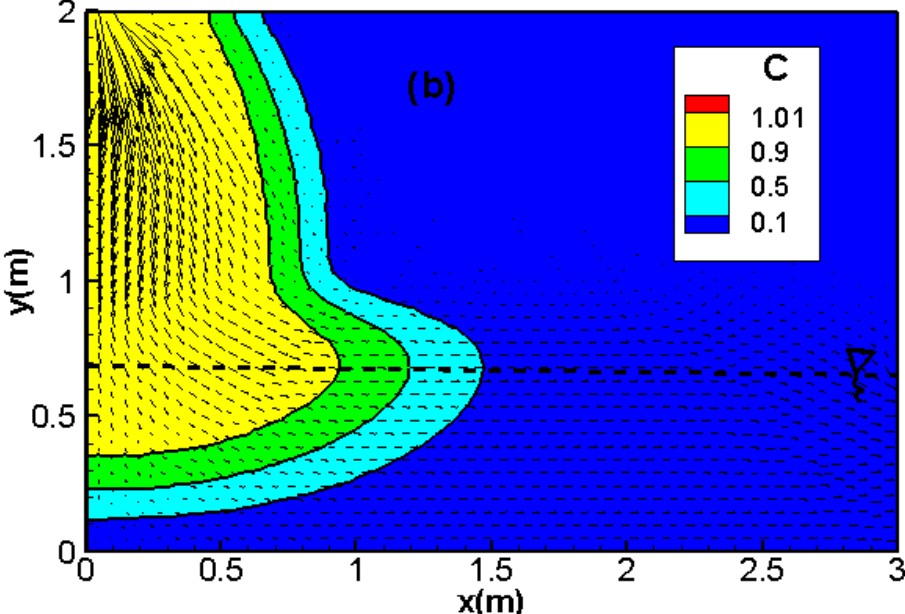


Figure 9: Concentration distribution with the hybrid-MFE (a) and upwind-MFE (b) methods

for the transport problem with low dispersion in variably-saturated porous medium.

## 6. Conclusion

MFE is a robust numerical method well adapted for diffusion problems on heterogeneous
domains and unstructured meshes. When applied to transport equations, the MFE solution can
exhibit strong unphysical oscillations due to the hyperbolic nature of advection. Upwind
schemes can be used to avoid such oscillations, although they introduce some numerical
diffusion. In this work, we developed an upwind scheme that does not require any
approximation for the upwind concentration. The method can be seen as a combination of an
upwind edge/face centred FV method with the lumped formulation of the hybrid-MFE
method. It ensures continuity of both advective and dispersive fluxes between adjacent
elements and allows to maintain the time derivative continuous, which facilitates employment
of high order time integration methods via the method of lines (MOL) for nonlinear problems.
Numerical simulations for the transport in a saturated porous medium show that the standard
hybrid-MFE method can generate unphysical oscillations due to the hyperbolic nature of
advection. These unphysical oscillations are completely avoided with the new upwind-MFE
scheme. The simulation of the problem of contaminant transport in a variably-saturated
porous medium shows that only the upwind-MFE scheme provides a stable solution. The
results point out the robustness of the developed upwind-MFE scheme when combined with
the MOL for solving nonlinear transport problems.



## Code/data availability

All data presented in this paper as well as the hybrid-MFE and the upwind-MFE Fortran transport codes are available under request from the first author.

## Author contributions

Anis Younes: conception and design of study, model development, drafting the manuscript

Hussein Hoteit: model verification, revising the manuscript

Rainer Helmig: writing and revising the manuscript

Marwan Fahs: design of study, literature review

## Competing interests

The authors declare that they have no competing interests

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

Methods Applied to Flow Problems in Porous Media, n.d.

Koohbor, B., Fahs, M., Hoteit, H., Doummar, J., Younes, A., and Belfort, B.: An advanced
discrete fracture model for variably saturated flow in fractured porous media, 140,
103602, https://doi.org/10.1016/j.advwatres.2020.103602, 2020.

Leij, F. J. and Dane, J. H.: Analytical solutions of the one-dimensional advection equation and
two-    or    three-dimensional    dispersion    equation,    26,    1475–1482,
https://doi.org/10.1029/WR026i007p01475, 1990.

Mazzia, A.: An analysis of monotonicity conditions in the mixed hybrid finite element
method    on    unstructured    triangulations,    76,    351–375,
https://doi.org/10.1002/nme.2330, 2008.

Mualem, Y.: A new model for predicting the hydraulic conductivity of unsaturated porous
media, Water Resour. Res., 12, 513–522, https://doi.org/10.1029/WR012i003p00513,
1976.

Putti, M., Yeh, W.W.-G., and Mulder, W.A.: A triangular finite volume approach with high-
resolution upwind terms for the solution of groundwater transport equations, Water
Resources Res., 26, 2865-2880, https://doi.org/10.1029/WR026i012p02865, 1990.

Radu, F. A., Suciu, N., Hoffmann, J., Vogel, A., Kolditz, O., Park, C.-H., and Attinger, S.:
Accuracy of numerical simulations of contaminant transport in heterogeneous
aquifers: A comparative study, Advances in Water Resources, 34, 47–61,
https://doi.org/10.1016/j.advwatres.2010.09.012, 2011.

Raviart, P. A. and Thomas, J. M.: A mixed finite element method for 2-nd order elliptic
problems, in: Mathematical Aspects of Finite Element Methods, Berlin, Heidelberg,
292–315, 1977.

Siegel, P., Mosé, R., Ackerer, P., and Jaffré, J.: Solution of the Advection Diffusion Equation
using a combination of Discontinuous and Mixed Finite Elements, Int. J. Numer.
Meth.    Fluids,    24:    595-613.    https://doi.org/10.1002/(SICI)1097-
0363(19970330)24:6<595::AID-FLD512>3.0.CO;2-I, 1997.


Traverso, L., Phillips, T. N., and Yang, Y.: Mixed finite element methods for groundwater flow in heterogeneous aquifers, Computers & Fluids, 88, 60–80, https://doi.org/10.1016/j.compfluid.2013.08.018, 2013a.

Traverso, L., Phillips, T. N., and Yang, Y.: Mixed finite element methods for groundwater flow in heterogeneous aquifers, Computers & Fluids, 88, 60–80, https://doi.org/10.1016/j.compfluid.2013.08.018, 2013b.

Vohralík, M.: A Posteriori Error Estimates for Lowest-Order Mixed Finite Element Discretizations of Convection-Diffusion-Reaction Equations, 45, 1570–1599, https://doi.org/10.1137/060653184, 2007.

Younes, A., Mose, R., Ackerer, P., and Chavent, G.: A New Formulation of the Mixed Finite Element Method for Solving Elliptic and Parabolic PDE with Triangular Elements, 149, 148–167, https://doi.org/10.1006/jcph.1998.6150, 1999.

Younes, A., Ackerer, P., and Lehmann, F.: A new mass lumping scheme for the mixed hybrid finite element method, International Journal for Numerical Methods in Engeneering, 67, 89–107, https://doi.org/10.1002/nme.1628, 2006.

Younes, A., Fahs, M., and Ahmed, S.: Solving density driven flow problems with efficient spatial discretizations and higher-order time integration methods, Advances in Water Resources, 32, 340–352, https://doi.org/10.1016/j.advwatres.2008.11.003, 2009.

Younes, A., Ackerer, P., and Delay, F.: Mixed finite elements for solving 2-D diffusion-type equations, Rev. Geophys., 48, RG1004, https://doi.org/10.1029/2008RG000277, 2010.

Younes, A., Koohbor, B., Belfort, B., Ackerer, P., Doummar, J., and Fahs, M.: Modeling variable-density flow in saturated-unsaturated porous media: An advanced numerical model, Advances in Water Resources, 159, https://doi.org/10.1016/j.advwatres.2021.104077, 2022a.

Younes, A., Hoteit H., Helmig, R., and Fahs, M.: A robust fully mixed finite element model for flow and transport in unsaturated fractured porous media, Advances in Water Resources, Volume 166, https://doi.org/10.1016/j.advwatres.2022.104259, 2022b.

# **Table Captions**


Table 1: Numerical results for the new upwind-hybrid MFE method.
Table 2: Parameters for the problem of infiltration into a 2D variably-saturated porous
medium.

# Figure Captions

Figure 1: Vectorial basis functions for the MFE method.
Figure 2: Continuity of concentration and total flux between adjacent elements with the
hybrid-MFE method.
Figure 3: The lumping region $R_i$ associated with the edge $i$, sharing the elements $E$ and $E'$
and formed by the two simplex regions $S_i^E$ and $S_i^{E'}$.
Figure 4: Description of the problem of the contamination of a 2D saturated porous medium.
Figure 5: Concentration distribution with the hybrid-MFE and the upwind-MFE methods for
the 2D saturated transport problem (only the region $x \leq 70\,\text{m}$ is depicted).
Figure 6: Concentration profiles at y = 20m (a) and x = 20 m (b) with the analytical, hybrid-
MFE and upwind-MFE solutions.
Figure 7: Description of the problem of contaminant infiltration into a 2D variably-saturated
porous medium.
Figure 8: Concentration distribution, with the hybrid-MFE (a) and the upwind-MFE (b)
schemes for the transport problem with high dispersion in a variably-saturated porous
medium.
Figure 9: Concentration distribution with the hybrid-MFE (a) and upwind-MFE (b) methods
for the transport problem with low dispersion in variably-saturated porous medium.