# Peer review of "A robust Upwind Mixed Hybrid Finite Element method for transport in variably saturated porous media Anis Younes1\*, Hussein Hoteit2, Rainer Helmig3, Marwan Fahs1 1 Institut Terre et Environnement de Strasbourg, Université de Strasbourg, CNRS, EN"

_Hydrology and Earth System Sciences, 2022_

## Referee Comment (RC2)

**Report on the paper**
**"A robust upwind mixed hybrid finite element method for transport in variably saturated porous media "**
**by A. Younes, H. Hoteit, R. Helmig and M. Fahs**

The paper under review presents a new method to obtain an upwind discretization of the advection–diffusion equation, using mixed hybrid finite element. This is a non–trivial problem, as one would like to obtain a stable ans monotone method, and because the requirement for a hybrid discretization means that the balance equation needs to be local to each element. This is difficult to reconcile with the need for upwinding so as to obtain a stable method. The author review an existing method, and propose a new variation. The proposed method is validated on two numerical examples.

In this reviewer's opinion, the main advantage of the proposed method would that it avoids the need to discretize in time to derive the hybrid discretization. As argued by the authors, this enables the use of a higher order, or variable time step, discretization.

However, the paper as it is currently written suffers from several deficiencies, some major, some minor, that I would like to see addressed before publication.

**Specific comments**

- I do not understand the "derivation" of the method that is given is Section 3 (starting on page 12). Why is it allowed, or useful, or even correct, to use a steady state form of the basic mass conservation equation (12) ? This approximation is used to obtain equation (24), and then again in equation (29).

- As a consequence of the previous point, why would the proposed method be consistent ? And why do the authors expect that the method is stable (or satisfies a maximum principle) ? I understand that the paper, or even the journal, is not a numerical analysis paper, but any discussion shedding some heuristic light in the two issues of consistency and stability would be welcome.

- If no analysis is possible, then the authors should at least include a numerical convergence study, say for example 1, to quantify the accuracy of the proposed method.

- For the two examples discussed in Section 4, the proposed method is compared to the basic hybrid MFE method (that is equation (18) of the paper). But this method is already known to be non-monotone. A more significant comparison would be with the previous upwind variant from the Radu et al. (2011) reference quoted (that is the method described on page 11). This would allow the authors to discuss the possible

pros and cons of their proposed method, in terms of accuracy, robustness and versatility. Even though I do not ask lightly that the authors do more numerical experiments, I feel this is really needed before the paper can be published.

- The paper under review has a significant overlap with the recently published paper [1] by the same authors. The authors should clarify the relationship between both papers.

**Technical corrections**

- The shortcut " $+\{\}'$ " used first in equation (27) and several times after that is very confusing. Does it mean "additional terms that the authors do not want to detail", or rather "the same as before, but for element $E'$ instead of $E$" ? I know the answer is the latter, but I wasn't sure the first time. This really should be clarified, preferably by writing the equation in full, at least by giving an explanation.

- The exposition of the model in Section 2 should follow a more logical order: I would suggest starting with equation (3) (with (4) and (5)), then give equation (1) to say where $\mathbf{q}$ comes from (and explain what $\theta$ is, then conclude with equation (6). The discretized equations such as (2) and (7) should only appear later.

- I think there is a confusion between $B$ and $B^{-1}$ on line 162. And the correct notation should be $\tilde{B}_{i,j}^{E,-1}$.

- In principle, the flux terms in equations (16) and later should have a superscript $n+1$ attached.

- The physical units in Table 1 should be written in an upright font (as in the text before).

- Example 1 has been used as a test case in the same context by (at least) two papers: [2] and [3], that should be cited.

**References**

[1] *A robust fully mixed finite element model for flow and transport in unsaturated fractured porous media*, Anis Younes, Hussein Hoteit, Rainer Helmig, Marwan Fahs, Advances in Water Resources, Volume 166, 2022, `https://doi.org/10.1016/j.advwatres.2022.104259`.

[2] *A triangular finite volume approach with high-resolution upwind terms for the solution of groundwater transport equations*, M. Putti, W. W.-G. Yeh and W. A. Mulder, Water Resources Res., 26, 2865–2880 (1990).

[3] *Solution of the Advection–Diffusion Equation Using a Combination of Discontinuous and Mixed Finite Elements*, Siegel, P., Mosé, R., Ackerer, P. and Jaffre, J., Int. J. Numer. Meth. Fluids, 24: 595-613 (1997). `https://doi.org/10.1002/(SICI)1097-0363(19970330)24:6<595::AID-FLD512>3.0.CO;2-I`.

---

## Author Comment (AC3)

**Manuscript Number: hess-2022-153**

**Title:** A robust Upwind Mixed Hybrid Finite Element method for transport in variably saturated porous media

RC2: 'Comment on hess-2022-153', Anonymous Referee #2, 11 Jul 2022 reply

The paper under review presents a new method to obtain an upwind discretization of the advection--diffusion equation, using mixed hybrid finite element. This is a non-trivial problem, as one would like to obtain a stable and monotone method, and because the requirement for a hybrid discretization means that the balance equation needs to be local to each element. This is difficult to reconcile with the need for upwinding so as to obtain a stable method. The author review an existing method, and propose a new variation. The proposed method is validated on two numerical examples.

In this reviewer's opinion, the main advantage of the proposed method would that it avoids the need to discretize in time to derive the hybrid discretization. As argued by the authors, this enables the use of a higher order, or variable time step, discretization.

However, the paper as it is currently written suffers from several deficiencies, some major, some minor, that I would like to see addressed before publication.

> **Answer:** We thank the Reviewer for the constructive comments which helped us to improve the quality of our manuscript. As detailed below, all suggestions are accounted for in the revised version.

Specific comments

- I do not understand the "derivation" of the method that is given is Section 3 (starting on page 12). Why is it allowed, or useful, or even correct, to use a steady state form of the basic mass conservation equation (12) ? This approximation is used to obtain equation (24), and then again in equation (29).

  > **Answer:** We agree and improve the presentation of the method in the revised version. More details are provided to explain how and why it is useful to use the steady state form of the mass conservation equation.

  > The main idea of the new method is to combine the upwind edge centered finite volume method with the lumped hybrid MFE method. A new section is added to better

explain the principles of the new method. Indeed, the standard hybrid MFE method is based on two stages: (*i*) the discretization of the transient mass conservation equation over the element E and (*ii*) the continuity of fluxes across the edge i sharing the elements E and E'. With the lumped formulation, the previous two stages are transformed to new ones as following: (*i*) the transient mass conservation over E is transformed to a steady state one and (*ii*) the continuity equation, which is seen as a steady state mass conservation equation at the edge level, is transformed to a transient equation. The new scheme keeps the time derivative continuous which allows the use of efficient high order temporal discretization methods via the MOL.

These points are better specified in the revised version

- As a consequence of the previous point, why would the proposed method be consistent ? And why do the authors expect that the method is stable (or satisfies a maximum principle) ? I understand that the paper, or even the journal, is not a numerical analysis paper, but any discussion shedding some heuristic light in the two issues of consistency and stability would be welcome.

   **Answer:** In the new method, the term of mass (with time derivative) has a contribution only on the diagonal terms of the final system matrix. This improves the monotonous character of the hybrid-MFE solution. For dispersive transport, in the case of an acute triangulation, the maximum principle is respected whatever the heterogeneity of the porous medium (Younes *et al*., 2006). In the case of transport by only advection, the system matrix with the new upwind hybrid MFE scheme is always an *M*-matrix (a non singular matrix with $m_{ii} > 0$, $m_{ij} \leq 0$). The *M*-matrix property insures the stability of the scheme since it guaranties the respect of the discrete maximum principle *i.e.* local maxima or minima will not appear in the $C$ solution in a domain without local sources or sinks.

   Note that in the case of 2D triangular elements, the lumped formulation is algebraically equivalent to the nonconforming Crouzeix-Raviart (Crouzeix and Raviart, 1973) finite element method (see Younes et al, 2008). The nonconforming Crouzeix-Raviart method uses the chapeau functions as basis functions to approximate the concentration, like the standard finite element method, but seed nodes are the midpoints of the edges.

These points will be specified in the revised version

- If no analysis is possible, then the authors should at least include a numerical convergence study, say for example 1, to quantify the accuracy of the proposed method.

  **Answer:** We agree and will present results of simulations for the first test case (with the analytical solution) using different mesh refinements in order to investigate the order of convergence in space and time of the new upwind-hybrid MFE method.

- For the two examples discussed in Section 4, the proposed method is compared to the basic hybrid MFE method (that is equation (18) of the paper). But this method is already known to be non-monotone. A more significant comparison would be with the previous upwind variant from the Radu et al. (2011) reference quoted (that is the method described on page 11). This would allow the authors to discuss the possible pros and cons of their proposed method, in terms of accuracy, robustness and versatility. Even though I do not ask lightly that the authors do more numerical experiments, I feel this is really needed before the paper can be published.

  **Answer:** In the revised version, we add numerical experiments which show that the new scheme has optimal first order convergence in time and space as the upwind scheme of Radu *et al*. (2011) and the non-hybrid upwind mixed method of Dawson (1978).

  Note that the upwind scheme of Radu *et al*. (2011) has good convergence properties but uses the approximation that the concentration at the edge $i$ is the average of the mean concentrations of the two adjacent elements sharing $i$. This can be a rough approximation since with the hybrid MFE method, the trace of concentration can be significantly different from the average of mean concentrations of adjacent cells, especially in the case of a heterogeneous domain where dispersion can vary with several orders of magnitude from element to element. Further, with this scheme one cannot ensure continuity of both advective and dispersive fluxes at the interface since the continuity is prescribed only for the total flux. These two drawbacks are avoided with the new developed scheme.

- The paper under review has a significant overlap with the recently published paper (see below) by the same authors. The authors should clarify the relationship between both papers.

  **Answer:** The aim of the present paper is (i) to develop a new upwind hybrid MFE scheme for advection-dispersion transport, (ii) to detail the main principles of the new scheme and show its benefits as compared to other upwind MFE schemes, (iii) to validate the method by comparison against an analytical solution, (iv) to investigate its order of convergence in time and space (in the revised version) and (v) to show its robustness when combined with the MOL for transport in unsaturated zones.

  The developed upwind scheme for transport has been recently successfully combined with the MFE method for fluid flow to simulate coupled flow and transport in unsaturated fractured porous media (Younes et al., 2022). The main objective of the paper of Younes *et al.* (2022) was to investigate the efficiency and robustness of a 1D-2D discrete fracture matrix (DFM) approach to model nonlinear flow and transport problems in fractured porous media with matrix-fracture and fracture-fracture fluid and mass exchanges.

These points will be specified in the revised version

*A robust fully mixed finite element model for flow and transport in unsaturated fractured porous media*, Anis Younes, Hussein Hoteit, Rainer Helmig, Marwan Fahs, Advances in Water Resources, Volume 166, 2022, https://doi.org/10.1016/j.advwatres.2022.104259.

**Technical corrections**

• The $\left\{ \ \right\}'$ used first in equation (27) and several times after that is very confusing. Does it mean "additional terms that the authors do not want to detail", or rather "the same as before, but for element E0 instead of E"? I know the answer is the latter, but I wasn't sure the first time. This really should be clarified, preferably by writing the equation in full, at least by giving an explanation.

  **Answer:** We agree and change the text accordingly

• The exposition of the model in Section 2 should follow a more logical order: I would suggest starting with equation (3) (with (4) and (5)), then give equation (1) to say where $q$ comes from (and explain what $\theta$ is, then conclude with equation (6). The discretized equations such as (2) and (7) should only appear later.

**Answer:** We agree and change the order of equations in the revised version

• I think there is a confusion between $B$ and $B^{-1}$ on line 162. And the correct notation should be $\tilde{B}_{i,j}^{E,-1}$

**Answer:** Corrected in the revised manuscript.

• In principle, the flux terms in equations (16) and later should have a superscript n + 1 attached.

**Answer:** We agree and specify in the text that the subscript n+1 is omitted to alleviate the notation

• The physical units in Table 1 should be written in an upright font (as in the text before).

**Answer:** Corrected in the revised manuscript.

• Example 1 has been used as a test case in the same context by (at least) two papers: [2] and [3], that should be cited.

**Answer:** The two papers are referenced in the revised version.

---

## Author Response (AR1)

**Manuscript Number: hess-2022-153**

**Title:** A robust Upwind Mixed Hybrid Finite Element method for transport in variably saturated porous media

**CC1: 'Comment on hess-2022-153', Thomas Graf, 25 May 2022**

This manuscript introduces a new upwind Mixed Finite Element (MFE) method to solve for solute transport under saturated and unsaturated conditions. The new scheme is shown to avoid unphysical oscillations that is otherwise potentially caused by advection.

The manuscript is clearly written, and the mathematical context is sound and complete. The two illustrative results are useful and demonstrate very well the capabilities of the new method.

> **Answer:** We highly appreciate the Reviewer's positive appraisal of our work as well as careful reading of the paper. As detailed below, all comments are accounted for in the new revised version.

There are some minor comments that I ask the authors to address:

1. There are a number of filling words (especially "indeed") that need to be removed. It is also uncommong to use "the" in e.g. line 129 when generally addressing water content etc.

> **Answer:** Corrected in the entire document.

2. Description of boundary conditions of the two examples in section 4. is not complete. Both figures 4 and 7 should show all BCs for both flow and transport. As is, this is not the case and must be changed. Also, the text does not give the full description of all BCs for both flow and transport, which also has to be completed.

> **Answer:** Figures 4 and 7 have been changed to include all flow and transport boundary conditions. A full description of all BCs is also given in the text of the revised version.

A figure that shows the meshes for both examples is missing and would be very helpful. I understand it is an unstructured mesh which explains the fact that result in Fig. 5 is not symmetric. This should be mentioned.

**Answer:** Although we have used an unstructured mesh, the results of Figure 5 are almost symmetrical, except the unphysical oscillations obtained with the standard MFE scheme.

Note that in the revised version, new simulations have been added for the first test problem with different uniform meshes to investigate time and space convergence of the new developed method.

For simplicity, it would have been more efficient to use only the upper half of the domain due to symmetry reasons. The authors should give a reason why they did not do so.

**Answer:** Yes, for efficiency reasons, it is better to simulate only the upper half of the domain because of symmetry, but we prefer to simulate the whole domain to have more representative results and to show the capability of the new upwind MFE model to reproduce symmetrical results.

**RC1**: 'Comment on hess-2022-153', Anonymous Referee #1, 07 Jul 2022 reply

This paper deals with 2D numerical simulations for a coupled system arising in flow and transport in heterogeneous media. The mathematical model under consideration is the flow and transport in variably saturated porous media using Richard's equation. A numerical scheme is developed for the discretization of this system by combining a mixed finite element method and a new upwind scheme for the convective term. 2D numerical results are presented to see the performance of the scheme for two tests for numerical simulation of contaminant transport into a variably saturated porous medium. The obtained results are satisfactory.

The subject is of interest and of current events. The authors made an interesting contribution for a difficult problem. The paper is well written and the results are of current interest. I deeply recommend the publication of this article.

> **Answer:** We thank the Reviewer for his/her positive appraisal of our work. As detailed below, all comments are accounted for in the new revision.

The authors should clarify the following points:

- '{ }' notation should be defined to avoid confusion.

> **Answer:** The symbol $\{\ \}'$ designates the contribution from the adjacent element $E'$. This has been specified in the revised version.

- The time discretization, the strategy used for the choice of the time step, the resolution of the nonlinear system and the linear systems should be specified.

> **Answer:** Done in the revised manuscript.

- To ensure reproducibility of the results of the two tests presented, all necessary data including discretization and solvers etc. should be specified.

> **Answer:** We agree and add all data in the revised version.

- It would also be interesting to give information about the environment in which the simulations were performed and the CPU times for each simulation.

> **Answer:** We agree and add information about the environment for the simulations and CPU times.

- Can you comment on the extension of this approach to the 3D problem?

**Answer:** The scheme can be seen as a combination of the upwind edge/face centred Finite Volume (FV) method with the lumped formulation of the hybrid MFE method and, as such, it can be easily extended to 3D problems.

RC2: 'Comment on hess-2022-153', Anonymous Referee #2, 11 Jul 2022 reply

The paper under review presents a new method to obtain an upwind discretization of the advection--diffusion equation, using mixed hybrid finite element. This is a non-trivial problem, as one would like to obtain a stable and monotone method, and because the requirement for a hybrid discretization means that the balance equation needs to be local to each element. This is difficult to reconcile with the need for upwinding so as to obtain a stable method. The author review an existing method, and propose a new variation. The proposed method is validated on two numerical examples.

In this reviewer's opinion, the main advantage of the proposed method would that it avoids the need to discretize in time to derive the hybrid discretization. As argued by the authors, this enables the use of a higher order, or variable time step, discretization.

However, the paper as it is currently written suffers from several deficiencies, some major, some minor, that I would like to see addressed before publication.

> **Answer:** We thank the Reviewer for the constructive comments which helped us to improve the quality of our manuscript. As detailed below, all suggestions are accounted for in the revised version.

Specific comments

- I do not understand the "derivation" of the method that is given is Section 3 (starting on page 12). Why is it allowed, or useful, or even correct, to use a steady state form of the basic mass conservation equation (12) ? This approximation is used to obtain equation (24), and then again in equation (29).

  > **Answer:** We agree and improve the presentation of the method in the revised version. More details are provided to explain how and why it is useful to use the steady state form of the mass conservation equation.

  > The main idea of the new method is to combine the upwind edge centered finite volume method with the lumped hybrid MFE method. A new section is added in the revised version, to better explain the main principles of the method. The standard hybrid MFE method is based on two stages: (*i*) the discretization of the transient mass conservation equation over an element E and (*ii*) the continuity of fluxes across the edge *i* sharing the elements $E$ and $E'$. With the lumped formulation, the previous two stages are transformed as following: (*i*) the transient mass conservation over $E$ is

transformed to a steady state one and (*ii*) the continuity equation, which is seen as a steady state mass conservation equation at the edge level, is transformed to a transient equation at the edge *i*. The new scheme has better monotonicity and keeps the time derivative continuous which allows the use of efficient high order time discretization methods via the MOL.

These points are specified in the revised version

- As a consequence of the previous point, why would the proposed method be consistent ? And why do the authors expect that the method is stable (or satisfies a maximum principle) ? I understand that the paper, or even the journal, is not a numerical analysis paper, but any discussion shedding some heuristic light in the two issues of consistency and stability would be welcome.

  **Answer:** In the new method, the term of mass (with time derivative) has a contribution only on the diagonal term of the final system matrix. This improves the monotonous character of the solution.

  For dispersive transport, the maximum principle is respected in the case of an acute triangulation whatever the heterogeneity of the porous medium (Younes *et al*., 2006). The lumped formulation is algebraically equivalent to the nonconforming Crouzeix-Raviart (Crouzeix and Raviart, 1973) finite element method (Younes *et al*., 2008). The nonconforming Crouzeix-Raviart method uses the chapeau functions as basis functions to approximate the concentration, like the standard finite element method, but seed nodes are the midpoints of the edges.

  In the case of advection transport, the system matrix with the new proposed scheme is always an *M*-matrix (a non singular matrix with $m_{ii} > 0$, $m_{ij} \leq 0$). This property insures the stability of the scheme since it guaranties the respect of the discrete maximum principle *i.e.* local maxima or minima will not appear in the solution in a domain without local sources or sinks.

These points are specified in the revised version

- If no analysis is possible, then the authors should at least include a numerical convergence study, say for example 1, to quantify the accuracy of the proposed method.

**Answer:** We agree and present results of simulations for the first test case (with the analytical solution) using different mesh refinements in order to investigate the order of convergence in space and time of the new upwind-hybrid MFE method.

- For the two examples discussed in Section 4, the proposed method is compared to the basic hybrid MFE method (that is equation (18) of the paper). But this method is already known to be non-monotone. A more significant comparison would be with the previous upwind variant from the Radu et al. (2011) reference quoted (that is the method described on page 11). This would allow the authors to discuss the possible pros and cons of their proposed method, in terms of accuracy, robustness and versatility. Even though I do not ask lightly that the authors do more numerical experiments, I feel this is really needed before the paper can be published.

**Answer:** In the revised version, we add numerical experiments which show that the new scheme has optimal first order convergence in time and space as the upwind scheme of Radu *et al*. (2011) and the non-hybrid upwind mixed method of Dawson (1978).

Note that the upwind scheme of Radu *et al*. (2011) has good convergence properties but assumes the concentration at the edge $i$ to be the average of the mean concentrations of the two adjacent elements sharing $i$. This can be a rough approximation since with the hybrid MFE method, the trace of concentration at the edge can be significantly different from the average of mean concentrations of adjacent cells, especially in the case of a heterogeneous domain where dispersion can vary with several orders of magnitude from element to element. Further, with this scheme one cannot ensure continuity of both advective and dispersive fluxes at the interface since the continuity is prescribed only for the total flux. These two drawbacks are avoided with the new developed scheme.

These points are better specified in the revised version

- The paper under review has a significant overlap with the recently published paper (see below) by the same authors. The authors should clarify the relationship between both papers.

    **Answer:** The aim of the present paper is (*i*) to develop a new upwind hybrid MFE scheme for advection-dispersion transport, (*ii*) to detail the main stages of the new scheme and show its benefit as compared to other upwind MFE schemes, (*iii*) to validate the method by comparison against an analytical solution, (*iv*) to investigate its order of convergence in time and space (added in the revised version, following to your suggestion) and (*v*) to show its robustness when combined with the MOL for transport in unsaturated zones.

    The new transport scheme has been recently successfully combined with the MFE method for fluid flow to simulate coupled flow and transport in unsaturated fractured porous media (Younes *et al*., 2022). The main objective of the paper of Younes *et al.* (2022) was to investigate the efficiency and robustness of a 1D-2D discrete fracture matrix (DFM) approach to model nonlinear flow and transport problems in fractured porous media.

These points are specified in the revised version

*A robust fully mixed finite element model for flow and transport in unsaturated fractured porous media*, Anis Younes, Hussein Hoteit, Rainer Helmig, Marwan Fahs, Advances in Water Resources, Volume 166, 2022, https://doi.org/10.1016/j.advwatres.2022.104259.

**Technical corrections**

• The $\{\ \}'$ used first in equation (27) and several times after that is very confusing. Does it mean "additional terms that the authors do not want to detail", or rather "the same as before, but for element E0 instead of E"? I know the answer is the latter, but I wasn't sure the first time. This really should be clarified, preferably by writing the equation in full, at least by giving an explanation.

    **Answer:** We agree, the text has been changed to clarify the notation

• The exposition of the model in Section 2 should follow a more logical order: I would suggest starting with equation (3) (with (4) and (5)), then give equation (1) to say where $q$

comes from (and explain what $\theta$ is, then conclude with equation (6). The discretized equations such as (2) and (7) should only appear later.

**Answer:** We agree and change the order of equations accordingly

• I think there is a confusion between $B$ and $B^{-1}$ on line 162. And the correct notation should be $\tilde{B}_{i,j}^{E,-1}$

**Answer:** Corrected in the revised manuscript.

• In principle, the flux terms in equations (16) and later should have a superscript n + 1 attached.

**Answer:** We agree and specify in the text that the subscript n+1 is omitted to alleviate the notation

• The physical units in Table 1 should be written in an upright font (as in the text before).

**Answer:** Corrected in the revised manuscript.

• Example 1 has been used as a test case in the same context by (at least) two papers: [2] and [3], that should be cited.

**Answer:** The two papers are referenced in the revised version.

---

## Author Response (AR2)

**Manuscript Number: hess-2022-153**

**Title:** A robust Upwind Mixed Hybrid Finite Element method for transport in variably saturated porous media

RC2: 'Comment on hess-2022-153', Anonymous Referee #2, 14 Sep 2022 reply

I thank the authors for taking into account my previous comments.

I still have one point that I think needs clarification. In the new 3.2.1 section, I still do not understand the derivation of the method: why is it possible to use equation (26) (with (28) as a result), and later (31) ? The flow is obviously not steady, so if this is an approximation, why is the error expected to be small? I could not even make the connection with the way the method is presented in Younes et al (2006) Equation (28) there is the same as (32) in the current paper, but the derivation given there is different.

Additionally, the authors might want to move Section 3.2.1 to a new Section 3.1.2, as a review of existing methods, and keep the new hybrid, lumped, upwind method to section 3.2. This is merely a suggestion.

> **Answer:** As detailed in the section 3.2.1, the standard hybrid-MFE method is based on the transient equation (24) at the element level and the steady state equation (25) at the edge level. With the lumped formulation, we switch and write the steady state equation at the element level (equation (26) and the transient equation at the edge level (equation (29)). For triangular meshes, the obtained scheme is algebraically equivalent to the nonconforming Crouzeix-Raviart (Crouzeix and Raviart, 1973) finite element method (Younes et al., 2008).
>
> We prefer to keep the sub-section 3.2.1 as a part of the section 3.2 since it explains the lumped formulation for the simple case of dispersion before introducing the new upwind scheme in 3.2.2.

---

## Author Response (AR3)

**Manuscript Number: hess-2022-153**

**Title:** A robust Upwind Mixed Hybrid Finite Element method for transport in variably saturated porous media

**Editor Comments**

I am sorry, but the referee's comment has not been carefully considered. I double checked the version of the manuscript which should contain tracked changes, but I could not find any change. I also compared the file with the previous version of the manuscript, but I was not able to identify any differences. So, it seems that no changes have been done. If this is the case, please, state it explicitly in your answer.

I have not fast, direct access to Younes et al. (2006) to check the differences, that were mentioned by the referee, between the derivation given in this manuscript and that in the 2006 paper. Nevertheless, if I remain sticked to the submitted manuscript, it is not clear how the steady-state dispersive flux is defined and how it is related to the "real" transient quantities. I think I "grasp" the idea, but I am not fully sure that I correctly understood the derivation. So, I think that it is fundamental to **slightly reformulate section 3.2.1, in order to clarify the mathematical arguments.**

Moreover, even if the referee stressed that the modification of the structure of section 3 was a suggestion, I think that **a simple reorganization could significantly improve the manuscript.** For instance, the new structure could be something like this:

3.1 Existing approaches to upwind-MFE

3.1.1. The upwind-hybrid MFE of Radu et al. (2011)

3.1.2 The lumped hybrid-MFE scheme for dispersion transport by Younes et al. (2006)

3.2 The new upwind lumped hybrid-MFE scheme for advection-dispersion transport

> **Answer:** We thank the reviewer and the editor for their suggestions to improve the quality of our manuscript. All the suggestions are accounted for in the new version.
>
> More explanations are added in the revised version for the new upwind lumped hybrid-MFE scheme. Further, the structure of the paper has been modified. The lumped scheme of Younes et al. (2006), developed for dispersion, does not contain any upwinding procedure and, as such, cannot be included in the upwinding section. Instead, in the revised version, the upwind-hybrid MFE of Radu et al. (2011) and the

lumped hybrid-MFE scheme of Younes et al. (2006) form a new section devoted to these two approaches which have been developed to improve the stability of the MFE solution of the transport equation.

The new structure of the paper is as following:

**1. Introduction**

**2. The hybrid-MFE method for the advection-dispersion equation**

**3. The upwind and lumped MFE approaches**

      3.1 The upwind-hybrid MFE of Radu et al. (2011)

      3.2 The lumped hybrid-MFE scheme for dispersion transport

**4. The new upwind-hybrid MFE scheme for advection-dispersion transport**

**5. Numerical Experiments**

      5.1 Transport in saturated porous media: comparison against a 2D analytical solution

      5.2 Transport in a variably-saturated porous medium

**6. Conclusion**